



# Implementation of a dry surface layer soil resistance in two contrasting semi-arid sites with SURFEX-ISBA V9.0

Belén Martí[1], Jannis Groh[2,3,4], Guylaine Canut[1], and Aaron Boone[1]

[1]CNRM, Université de Toulouse, Météo-France, CNRS, Toulouse, France
[2]Institute of Crop Science and Resource Conservations (INRES), Soil Science and Soil Ecology, University of Bonn, Bonn, Germany
[3]Institute of Bio- and Geosciences - Agrosphere (IBG-3), Forschungszentrum Jülich GmbH, Jülich, Germany
[4]Research Area 1 Landscape Functioning, Isotope Biogeochemistry and Gas Fluxes, Leibniz Centre for Agricultural Landscape Research (ZALF), Müncheberg, Germany

**Correspondence:** Belén Martí (belen.marti@meteo.fr)

**Abstract.** The estimation of latent heat fluxes in semi-arid regions faces several challenges, such as human intervention in the water cycle through irrigation, sudden changes in vegetation state due to crop harvesting, the still evolving knowledge of the physical processes governing plant transpiration and soil evaporation, and the lack of measurements to develop and test models. Representing the wide range of evapotranspiration values presents difficulties for both simulations and measurements, owing to strong soil/plant spatial heterogeneity at relatively small (e.g. hectometric) scales. The ability to accurately predict the partition of evapotranspiration into evaporation and transpiration from observation is still very limited, but improved estimates are required so that better water use decisions can be made. Land surface models (LSMs) can be used as a tool in this regard, when their validation is possible, simulations tend to overestimate soil evaporation in most models.

The simulations in this study make use of the LSM ISBA, which represents the land component within the surface coupling platform SURFEX. They include two field sites with contrasting soil moisture and vegetation characteristics during the summer of the Land surface interactions with the atmosphere over the Iberian semi-arid environment (LIAISE) campaign. The first site corresponds to a full cutting and growing cycle of one month in a flood irrigated alfalfa field. A detailed examination of the parametrization suggests that several parameters determine the amount and tendency of transpiration change. In particular, a higher quantum efficiency and maximum assimilation are marked as the driving model parameters together with a mesophylic conductance value closer to C4 behavior. The second site is an uncultivated rain-fed area of natural grass close to senescence. As the parametrization of the vegetation proved to be insufficient to characterize the evapotranspiration, for this study the implementation of a dry surface layer (DSL) resistance within the LSM ISBA was developed. The consideration of this process characterizes the transfer of vapor in a physical way that has proved successful in improving the partitioning of evapotranspiration in other models. The implementation of a DSL resistance led to an improvement in the simulated latent heat flux by reducing bare soil evaporation compared to simulations without a soil resistance. This approach resulted in a reduction in daily latent heat flux RMSE of 29% and 32% for the alfalfa and natural grass site respectively, while increasing slightly the correlation by 0.02 and 0.01 at both sites. Sensible heat flux and net radiation are improved on the order of $10\,\mathrm{W\,m^{-2}}$ whereas the ground heat flux is deteriorated within the same order. The resulting DSL simulations reduced the overall global error





compared to a simulation without a DSL resistance. Sensitivity tests of the parameters that drive a DSL resistance in ISBA further improve the simulations, reducing excessive damping after rain events. The new DSL parameterization helps overcome current problems of ET modelling by reducing bare soil evaporation within LSMs.

**Short Summary**. The characterization of vegetation at two sites proved insufficient to simulate adequately the evapotranspiration. A dry surface layer was implemented in the land surface model SURFEX-ISBA v9.0. It is compared to simulations without a soil resistance. The application to an alfalfa site and a natural grass site in semiarid conditions results in an improvement in the estimation of the latent heat flux. The surface energy budget and the soil and vegetation characteristics are explored in detail.

## 1 Introduction

Semi-arid environments are characterized by water deficits during part of the year. In these areas, water resource management is essential for food production and water use for the local population, as water scarcity is common. In particular, the estimation of water loss by evapotranspiration (ET) is a challenge in these areas due to several factors, such as the significant heterogeneity of land surface characteristics, low values of fluxes at the limits of observational capabilities (Mauder et al. 2020), anthropic changes in land use such as the harvesting of crops and in land management such as irrigation, and complex underrepresented processes in the unsaturated zone of the soil (e.g. secondary drying fronts, pore scale influence such as in Or et al. 2013). As a result, such processes are still being studied and incorporated to the models. Global estimates of ET reveal seasonal differences depending on the product (Jiménez et al. 2011) and the method (Coenders-Gerrits et al. 2014; Schlesinger and Jasechko 2014). The transpiration ratio, the proportion of water vapor released through transpiration relative to total ET, is critical in assessing the potential improvement of water use through the regulation of transpiration (Yoo et al. 2009). Estimates of the ratio range from transpiration-dominated (Miralles et al. 2011) to evaporation-dominated (Alton et al. 2009) ecosystems when estimated from satellite and model-derived ET. The observed range of transpiration ratios, from 38% to 77%, can largely be attributed to differences in leaf area index ($LAI$) estimates across the reviewed studies (Wang et al. 2014). Since LSMs tend to have a low transpiration ratio (Chang et al. 2018) and tend to overestimate the evaporative fraction and the bare soil evaporation after rain events (Lohou et al. 2014), a better representation of soil resistances can serve to improve the soil evaporation in the models, leading to better partitioning of ET and a reduction in near surface temperature biases in atmospheric models (Dong et al. 2020).

LSMs estimate ET and its partitioning with varying complexity (Noilhan and Planton 1989; Sellers et al. 1996; Coudert et al. 2006; Napoly et al. 2017). ET estimates can be made without distinguishing between sources, as in the Priestley-Taylor or Penman-Monteith methods, or by partitioning ET into bare soil evaporation, transpiration and evaporation from the interception of residual water on the leaves using similarity theory. Transpiration is modeled using a resistance analogue, with a canopy resistance which represents plant biophysical processes which can include the assimilation of carbon, the latter allows to modify the parameters to a particular plant species.



Bare soil evaporation in the unsaturated zone is characterized by the simultaneous presence of water in both liquid and vapor phases within the soil. Explicit treatment of soil water vapor requires full coupling between the transfer of heat and the transport of water (Philip and De Vries 1957; Milly 1984), which is very sensitive to soil hydraulic properties (Chanzy et al.

2008; Merlin et al. 2016). They can modify the behavior of capillary continuity and exchange processes from pores through air to the atmosphere, determining transport processes across soil drying stages (Or et al. 2013). Up to four stages of drying soil can be identified (Zhang et al. 2015): a first stage of ponding, a period of air intrusion into the soil, a rapid drying until no liquid remains and a residual layer drying as the vaporization plane descends and a dry surface layer (DSL) forms (Balugani et al. 2018). A more common approach is to separate soil evaporation in two stages. Stage 1 depends mostly on the atmospheric

energy demand and drives a constant rate of evaporation. Stage 2 evaporation is controlled by the soil characteristics. The rate of evaporation depends on the type of soil and its soil hydraulic properties (Schneider et al. 2021). The implicit treatment of bare soil evaporation can be modeled by a soil resistance thereby simplifying the coupling that depends on radiation, liquid water gradients and temperature gradients to a dependence on water content, and in some cases, also on the temperature.

Several empirical soil resistance approaches have been developed since the nineties, as simulated latent heat fluxes (espe-

cially from the soil) were found to be overestimated in their model (Sellers et al. 1992b), and more recently more physically meaningful resistances have been parameterized (Swenson and Lawrence 2014) (see more in Sect. 2.3.1). The aforementioned study used the FLUXNET dataset (Jung et al. 2009) and a model tree ensemble approach to analyze behavior on a global scale. By implementing the DSL in the Interactions Soil-Biosphere-Atmosphere (ISBA) model (see Sect. 2) , we investigate whether it can improve its estimation of bare soil evaporation and consequently ET.

Both parametrizations are tested in detail for two sites that were operated during the *Land surface interactions with the atmosphere over the Iberian Semi-arid environment* (LIAISE) campaign, described in Sect. 3.1. They represent the extremes found in the semi-arid environment in terms of Bowen ratio: the first site is a flood irrigated alfalfa field where ET dominates, and the second is an almost senescent rainfed natural grass site where sensible heat flux dominates. Together these two sites provide a good test case for studying the limits of the model and for evaluating the default and new parametrizations with a

DSL.

## 2    Model description

ISBA includes a Multi-Energy Budget (MEB) model option described in Boone et al. (2017) which uses a classical Big-Leaf type approach to modeling the SEB. MEB separates ET as the contribution of soil evaporation, transpiration, interception of evaporation and sublimation when snow is present. It allows separate consideration of vegetation-driven processes, such as

the impact of stomatal conductance and light assimilation on transpiration, and of soil processes, such as soil evaporation and the impacts of a litter layer. MEB can also represent vertical heterogeneity of the soil hydraulic and thermal properties across the soil profile. ISBA-MEB within SURFEX (*Surface Externalisée*: Externalized Surface) version V9 is used in this article together with the multilayer diffusive soil scheme option (Decharme et al. 2011). To date, the local scale evaluation of ISBA-



MEB has been carried out with trees (Napoly et al. 2017), corn (Dare-Idowu et al. 2021) and in a semi-arid environment in a
vineyard (Aouade et al. 2020).

## 2.1 Components of the latent heat flux

The latent heat flux ($LE$) can be found by the evaporative contribution of the vegetation and the soil:

$$LE = L_v \left( E_v + E_g \right) \tag{1}$$

where $L_v$ is the latent heat of evaporation and $E_g$ is the part of the flux originated from soil evaporation. It is described in
detail subsection 2.3. $E_v$ is given by the sum of the plant transpiration $E_{tr}$ and the evaporation from the canopy liquid water
interception store $E_r$, when no snow is present. They are given by:

$$E_{tr} = \rho_a \frac{q_{sat,v} - q_c}{R_{avg-c} + R_s} (1 - \delta) \tag{2}$$

and

$$E_r = \rho_a \frac{q_{sat,v} - q_c}{R_{avg-c}} \delta \tag{3}$$

with $R_s = 1/g_{sI}$ where $g_{sI}$ is the stomatal conductance at the canopy level and it is the weighted average of the different plant
types multiplied by the $LAI$, $\rho_a$ is the air density, $q_{sat,v}$ is the saturated specific content at the vegetation temperature and $q_c$
is the canopy air specific humidity ($kgkg^{-1}$). $R_{avg-c}$ is the resistance between the overlaying air and vegetation. It consists on
the inverse of the sum of the bulk canopy aerodynamic conductance between the canopy and the canopy air (Choudhury and
Monteith 1988) and the conductance accounting for the free convection from (Sellers et al. 1986). $\delta$ is the Halstead coefficient
with a factor to account that saturated vegetation can transpire. See Boone et al. (2017) section 2.6.1 for the particulars of the
aerodynamic resistance of the canopy, section 2.8.3 for the Halstead coefficient and their appendix C2 for the $LE$ formulation,
which fall outside the discussions of this article.

## 2.2 Transpiration formulation: A-gs

The A-gs (Assimilation-stomatal conductance) scheme of Goudriaan et al. (1985) within SURFEX presents the parametrization
of plant assimilation of carbon. It was modified by Jacobs et al. (1996) to model photosynthesis and it can be used to obtain
transpiration. Different types of vegetation can be selected and its parametrizations are modeled as a function of the so-called
vegetation patch (or land cover type). In SURFEX, a patch corresponds to a Plant Functional type (PFT), which characterizes
subcategories natural land-surfaces. Additionally, vegetation can have either drought tolerant or drought avoidant strategies
(Calvet et al. 2004). A description of the processes can be found in Calvet et al. (1998) and Delire et al. (2020).

In order to discuss the parameter selection, we provide the mathematical development of the low vegetation scheme in ISBA
(Calvet 2000), which is used to represent crops. Transpiration in A-gs is given by:

$$E_T = \rho_a g_s D_s \tag{4}$$





**Table 1.** Table of default parametrized values in the A-gs scheme for the used PFT.

|  | f | $f_0^*$ | $D_{max}^N$ | $D_{max}^X$ | $f_{2c}$ | $Q_{10}(25)$ | $T_1(g_m^*, A_{m,max})$ | $T_2(g_m^*, A_{m,max})$ |
|---|---|---|---|---|---|---|---|---|
| units | m$^2$ kg$^{-1}$ | - | $g\,kg^{-1}$ | $g\,kg^{-1}$ | - | - | $^\circ C$ | $^\circ C$ |
| C3 herbaceous | 6.73 | 0.25 | 30 | 300 | 0.3 | 2.0 | 5,13 | 36,36 |
| C4 crop | -4.33 | 0.15 | 30 | 300 | 0.3 | 2.0 | 8,13 | 38,38 |

$D_s$ is the specific humidity deficit. $D_s$ indicates when stomata are closed by exceeding $D_{max}$, the maximum specific humidity deficit of the air tolerated by the vegetation (with no soil water stress). $D_s$ is of the form:

$$D_s = \frac{f - f_0^*}{f_{min} - f_0^*} D_{max} \tag{5}$$

where $f$ indicates the proportionality of internal $CO_2$ and inside the leaf boundary layer and $f_0^*$ is $f$ with no saturation deficit (without soil water stress) and are model parameters (see Table 1). $f_{min}$ is the minimal value of $f$:

$$f_{min} = \frac{g_c}{g_c + g_m^*} \tag{6}$$

where $g_m^*$ is the mesophyllic conductance and $g_c$ is the cuticular conductance. $g_m^*$ determines the gas exchange amount through the stomata and $g_c$ through the cuticle when stomata are completely closed. The cuticle is a kind of wax membrane that protects the leaf and allows little exchange of gases, its values being smaller than those of $g_m^*$. Both contribute to the total transpiration.

### 2.2.1 Evaporation strategies

$D_{max}$ behavior varies with evaporation strategy, limiting transpiration depending on its value. The formulation is the following:

$$D_{max} = D_{max}^+ + (D_{max}^* - D_{max}^+)\frac{f_2 - f_{2c}}{1 - f_{2c}} \tag{7}$$

where $D_{max}^+$ can be N for the drought-avoiding strategy applied for C3 crops, and X for the drought-tolerant strategy. For C3 crops $D_{max}^*$ is given by:

$$\ln(g_m^*) = 2.381 - 0.6103 \ln(D_{max}^*) \tag{8}$$

and

$$f_2 = \sum_{i=1}^{Ng} \zeta_i \frac{\omega_{g,i} - \omega_{wilt,i}}{\omega_{fc,i} - \omega_{wilt,i}} \tag{9}$$

where $f_2$ is the normalized soil moisture and $f_{2c}$ is its tabulated critical value. $f_2$ is summed over the soil layers $Ng$ (Decharme et al. 2011). Soil water content variables are represented by $\omega$. The first subindex indicates the water content variable, $g$ is the ground, $wilt$ is the wilting point and $fc$ is the field capacity. The second subindex corresponds to the layer number to which it pertains, 1 would indicate the most superficial layer. $\zeta_i$ represents the difference in the cumulative root zone fraction for a given




layer. Within this strategy $D_{max}$ increases until critical soil moisture is reached ($f_2 = f_{2c}$), then the previous $g_m^*$ symbolized by $g_m^{*X}$ is updated and decreases with the water deficit: $g_m^* = g_m^{*X} f_2/f_{2c}$.

For the drought tolerant strategy used for C4 crops and in our simulated alfalfa, we take Eq. (7), with $D_{max}^*$ given by:

$$\ln(g_m^*) = 5.323 - 0.8923\ln(D_{max}^*) \tag{10}$$

In this strategy then $D_{max}$ increases until the critical soil moisture is reached, $D_{max}$ decreases with the severity of the stress: $D_{max} = D_{max}^X f_2/f_{2c}$. Note that this is the opposite behavior as $D_{max}$ and $g_m^*$ are anticorrelated.

### 2.2.2 Stomatal conductance

To obtain $g_s$ we use the following expressions:

$$g_s = 1.6g_{sc} + g_c \tag{11}$$

where $g_{sc}$ is the stomatal conductance to $CO_2$ given by:

$$g_{sc} = g_{sc}^{first} + E^{first} \frac{M_a}{\rho_a M_v} \frac{C_s + C_i}{2(C_s - C_i)} \tag{12}$$

where $M_a$ and $M_v$ are molecular masses of air and water vapor respectively, $C_s$ is the external concentration of $CO_2$ and $C_i$ the $CO_2$ internal concentration, expressed as:

$$C_i = fC_s + (1+f)\Gamma \tag{13}$$

and

$$\Gamma(Ts) = \Gamma(25)Q_{10}^{Ts-25}/10 \tag{14}$$

where $\Gamma$ is the $CO_2$ compensation concentration at the skin temperature, $Ts$ is the superficial temperature of the leaf and $Q_{10}$ is fixed at 2.0 (see Table 1).

To obtain $E^{first}$ with equation (4), $g_s^{first} = 1.6g_{sc}^{first}$ accounting for the ratio of $CO_2$ to water assimilation for the stomatal conductance to $CO_2$ with:

$$g_{sc}^{first} = \frac{A_n - A_{min}\left[\frac{D_s}{D_{max}^*}\left(\frac{A_n+R_d}{A_m+R_d}\right) + R_d\left(1 - \frac{A_n+R_d}{A_m+Rd}\right)\right]}{C_s - Ci} \tag{15}$$

where $R_d = A_m/9$ and corresponds to the dark respiration. $A_m$ is the $CO_2$ assimilation limited by the air $CO_2$ concentration due to saturation, and $A_n$ is the $CO_2$ assimilation limited by the air $CO_2$ concentration. $A_{min}$ represents the residual photosynthesis rate (at full light intensity) associated with cuticular transfers when the stomata are closed because of a high specific humidity deficit, it is expressed as:

$$A_{min} = g_m^*(C_{min}\Gamma) \tag{16}$$





where $C_{min}$ is the minimum value of $C_i$ at $D^*_{max}$, given by:

$$C_{min} = \frac{g_c C_s + g^*_m \Gamma}{g_c + g^*_m} \tag{17}$$

Additionally, $A_m$ is:

$$A_m = A_{m,max}\left(1 - \exp\frac{-g^*_m(C_i - \Gamma)}{A_{m,max}}\right) \tag{18}$$

with:

$$A_{m,max} = \frac{A_{m,max}(25)Q_{10}^{(Ts-25)/10}}{(1 + \exp\left[0.3(T_1 - T_s)\right])(1 + \exp\left[0.3(T_s - T_2)\right])} \tag{19}$$

with $T_1$ and $T_2$ being reference values.

$$A_n = (A_m + R_d)\left(1 - \exp\left(\frac{-\epsilon I_a}{A_m + R_d}\right)\right) - R_d \tag{20}$$

where

$$\epsilon = \epsilon_0\left(\frac{C_i - \Gamma}{C_i + 2\Gamma}\right) \tag{21}$$

where $\epsilon_0$ is the quantum efficiency.

The relationships from (4) to (21) show that $g_s$ is highly nonlinear with the parameters $g^*_m$, $g_c$, $A_{max}$ and $\epsilon_0$. $D_s$ is in fact $D_s(g^*_m, g_c)$ and $A_n = A_n(g^*_m, g_c, \epsilon_0, A_{max})$ between others. Discussion on the values of these parameters is done in Sect. 4.2.1.

### 2.3 Bare soil model formulation

The SURFEX V9 MEB bare soil evaporation with soil resistance consists in a mixed form soil resistance such as in Niu et al. (2011), Xue et al. (1996) and Sellers et al. (1996). A mixed formulation consists in adding a soil resistance (beta type formulation) to the aerodynamic resistance while also using a soil humidity factor applied to the saturated specific humidity in the numerator (alpha type formulation). This added mixed formulation is incorporated in the ground evaporation expression as:

$$E_g = \rho_a\left(\frac{h_u q_{satg} - q_c}{R_{ag} + R_{soil}}\right) \tag{22}$$

$R_{ag}$ is the air resistance (s m$^{-1}$), $R_{soil}$ is the soil resistance, $q_{satg}$ is the saturated specific humidity of the air calculated with temperature of the ground at its first layer and $h_u$ is the soil humidity coefficient which has the form:

$$h_u = \begin{cases} \frac{1}{2}\left[1 - \cos\left(\frac{w_{g,1}}{w_{fc,1}}\right)\right], & w_{g,1} < w_{fc,1} \\ 1, & w_{g,1} \geq w_{fc,1} \end{cases} \tag{23}$$

and when $h_u * q_{satg} < q_c$ and $q_{satg} > q_c$ then either there's no soil evaporation because the low level humidity is dry ($Eg = 0$) or $h_u * q_{satg} < q_c$ and $q_{satg} \leq q_c$ which implies condensation (Noilhan and Mahfouf 1996). The default version of ISBA only included the humidity factor, $h_u$ (and thus was uniquely an alpha formulation).



### 2.3.1 Soil resistances

Overestimation of bare soil evaporation is generally present in LSMs (Chang et al. 2018), leading to overestimation of global ET (Wang et al. 2021). The partitioning of ET into transpiration and soil evaporation is the main source of inter-model differences for simulated ET amongst different models (Feng et al. 2023). Empirically-based soil resistances have been used for years in various models (Takata et al. 2003; Best et al. 2011; Harris et al. 2017; Raoult et al. 2018) with tuned values that can have a large impact on water balance components, such as runoff (Cuntz et al. 2016).

Several forms of soil resistance have been proposed over the years (Barton 1979; De Silans et al. 1989; Sellers et al. 1992b; Van de Griend and Owe 1994; Xue et al. 1996; Camillo and Gurney 1986; Mohamed et al. 1997; Ding et al. 2015; Ivanov et al. 2008), some of which have been tested within ISBA (Béziat et al. 2013). These were based on identifiable changes in the soil. De Silans et al. (1989) reported that the influence of soil resistance on evaporation was significant and goes along with a change in soil color together with a change in albedo measurements. Van de Griend and Owe (1994) observed the superficial DSL color change to a soil depth of 2.5 cm, together with a lower volumetric water content compared to the underlying layer and an increased estimation of the soil resistance value.

More recently, models have applied more data-driven approaches (Merlin et al. 2016; Lehmann et al. 2018; Raoult et al. 2021) to constrain case-dependent results and physically based resistances (Sakaguchi and Zeng 2009; Zhang et al. 2015; Swenson and Lawrence 2014) as knowledge of soil physics advances. These resistances differ in the value of the volumetric water content ($VWC$) at which they become active, whether its form is exponential or linear, and in terms of the magnitude of the resistance. The formulations cited here all produce lower resistance values than Sellers et al. (1992a) except for the DSL resistance as implemented by Swenson and Lawrence (2014). A comparison of resistance values for several standard soil resistance formulations over a range of soil water contents is shown in Fig. 6 of Swenson and Lawrence (2014), which shows that the DSL approach has the largest values. The DSL resistance was tested successfully globally in Swenson and Lawrence (2014). In the current study, we investigate this method in detail using the LIAISE data.

The soil drying process has been explored in laboratory conditions (Or et al. 2013; Zhang et al. 2015; Wang 2015; Merz et al. 2016; Balugani et al. 2021). The results of these studies have led to a better understanding of the process of DSL formation and its impact on ET. (Iden et al. 2021) has shown that including processes such as corner and film flow allows the identification of soil hydraulic properties with models such as HYDRUS in medium to dry conditions.

### 2.3.2 Sellers, 92

A widely available expression among LSMs such as community Noah-MP (He et al. 2023), PX LSM WRF/CMAQ (Ran et al. 2016), ISBA-MEB (Boone et al. 2017), ORCHIDEE (MacBean et al. 2020) for the soil resistance originated from Sellers et al. (1992b, 1996) and is formulated as:

$$R_{soil} = \exp\left[ A - B\left( \frac{w_{g,1}}{w_{sat,1}} \right) \right] \tag{24}$$





where $A$=8.206 and $B$=4.255 (Sellers et al. 1992b). These values were computed using field measurements taken during the FIFE 89 campaign in Kansas (Sellers et al. 1992a) by inverting the SiB model (Sellers et al. 1992b) and finding the best fit for several sites in the area.

### 2.3.3 Dry soil layer resistance

A DSL resistance is tested in SURFEX as an alternative for a soil resistance to Sellers 92. It models sites where compaction and very intense heat cause all liquid water to be lost in the first few centimeters. This results in the formation of a DSL that makes evaporation difficult: this impediment to evaporation is due to the transport of water being done only by vapor water diffusion. This process is modeled in a pragmatic manner by using a surface layer resistance. According to Swenson and Lawrence (2014), a DSL can be parameterized by the equation:

$$\Delta DSL = \begin{cases} \Delta z_{dsl} \left( w_{dsl0} - w_{g,1} \right) / \left( w_{dsl0} - w_{air} \right), & w_{g,1} < w_{dsl0} \\ 0, & w_{g,1} \geq w_{dsl0} \end{cases} \tag{25}$$

where $\Delta z_{dsl}$ is the length scale of the maximum $\Delta DSL$ thickness (m) and is given a value of $0.015$ m as in Swenson and Lawrence (2014), and $w_{dsl0}$ is the moisture value at which the DSL becomes active. $w_{dsl0}$ depends on the porosity ($\Phi$) from $w_{dsl0} = K_{dsl}\Phi$ where $K_{dsl} = 0.8$ is the value found by Swenson and Lawrence (2014) which improves their ET estimation in semi-arid conditions. The porosity is defined as the saturated volumetric water content ($w_{sat,1}$) of the first layer, as this corresponds to the pore space available for water. $w_{g,1}$ corresponds to the moisture value at the top soil layer of the model, while $w_{air}$ is the "air dry" soil moisture defined as

$$w_{air} = w_{sat,1} \left( \frac{\Psi_{sat}}{\Psi_{air}} \right)^{1/b} \tag{26}$$

where the air dry matric potential is $\Psi_{air} = -10^4$ m, the saturated matric potential is represented by $\Psi_{sat}$ and the slope of the soil water retention curve is $b$. The soil resistance is expressed as

$$Rsoil = \frac{\Delta DSL}{D_{\nu a} \tau_v} \tag{27}$$

where $\tau_v$ is the tortuosity of the vapor flow paths through the soil matrix and $D_{\nu a}$ is the molecular diffusivity of water vapor flow in air (m$^2$ s$^{-1}$). The expression of $D_{\nu a}$ differs slightly from that used in Swenson and Lawrence (2014) through the dependence on pressure, $p$, and the exponent of temperature:

$$D_{\nu a} = 2.17 * 10^{-5} \left( \frac{p_0}{p} \right) \left( \frac{T_{g,1}}{T_f} \right)^{1.88} \tag{28}$$

where $p_0$ is a standard reference pressure (1000 hPa), $T_{g,1}$ being the first level of soil temperature and $T_f$ is the freezing temperature of water.

The tortuosity is then given by:

$$\tau_v = \Phi_{air}^2 \left( \frac{\Phi_{air}}{\Phi} \right)^{3/b} \tag{29}$$

where $\Phi_{air} = \Phi - w_{air}$ and represents the air-filled pore space.



## 3 Data and sites

### 3.1 The LIAISE campaign

The LIAISE campaign (Boone et al. 2025) was designed to improve the understanding of the impact of anthropization on the water cycle in semi-arid environments, with a particular focus on identifying the limitations of LSMs under these conditions. The field experiment took place in the north-eastern part of the Iberian Peninsula from April 2021 to the end of September 2021 (the Long Observational Period, LOP). Surface energy budget (SEB) stations were installed over alfalfa, maize, irrigated grass, vineyard, apple orchard, almond orchard and natural rainfed grass (Price 2023). An intensive Special Observation Period (SOP) took place during July 15-29 (Boone et al. 2025), with in-situ measurements of soil and vegetation properties and of the atmospheric boundary layer (ABL) up to the entrainment zone through a multi-institutional collaboration (Brooke et al. 2023). The region is characterized by an irrigated area with fruit trees, maize and alfalfa, and a rainfed area with wheat, olives, almonds and natural grassland. The two areas are separated by the Canal d'Urgell. This configuration creates a profound contrast in the SEB components between the areas. The two main sites of the campaign, La Cendrosa and Els Plans, included a 50 m tower, SEB and meteorological measurements, radio soundings, $LAI$ and vegetation height observations (Boone et al. 2025).

### 3.2 Irrigated alfalfa site: La Cendrosa

The La Cendrosa site is located within an alfalfa field that was irrigated by gravity flooding approximately every ten days during the growing season and periodically cut for harvest. A cycle of cutting and growing during the month of July 2021, characterized by in-situ values of $LAI$ and vegetation height, is prescribed as input (Fig. 1) for ISBA. The period starts with fully grown vegetation, but after a few days, the vegetation is cut. Growth starts rapidly after the next irrigation period and the vegetation height increases from about 10 cm to its maximum height of approximately 70 cm in 17 days. The $LAI$ increases accordingly (from 0.3 to 3) over the same period.

This rapid vegetation evolution has an impact on the observed fluxes (Fig. 2). Initially, the net radiation ($Rn$) starts at values near 680 W m$^{-2}$ and decreases a few days after mowing. However, the change is not immediately apparent in the measurements as it takes several days for the harvested alfalfa to be removed from the soil as it is left to dry in the field. After the initial growth period, when $LAI$ is sufficient to cover most of the soil, the $Rn$ returns to values close to 600 W m$^{-2}$. These observed values are not as high as before harvest because the water content is lower than after irrigation and this results in a higher albedo (see Sect. 5.3). $LE$ is the dominant flux at this site, and the ground heat flux ($G$) and the sensible heat flux ($H$) represent a small part of the energy balance, except for the period after cutting and at the beginning of the growing period. The energy balance residue ($Res$) can be 100 W m$^{-2}$ and negative during the day. It is defined as the residual available energy remaining after the $Rn$ has been redistributed in the atmospheric and ground heat fluxes. The highest values of the residual are observed during periods of high vegetation. When vegetation is low, the value of the residue remains low or negative.

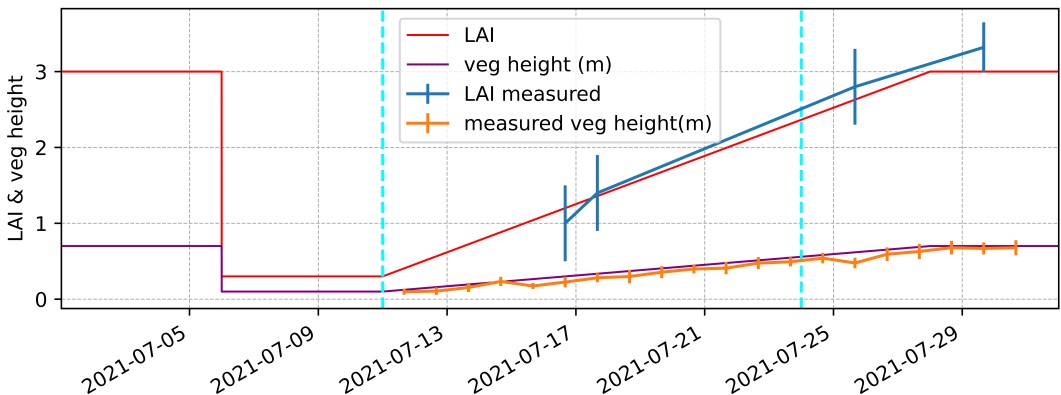

**Figure 1.** Cycle of cutting and growth of alfalfa at La Cendrosa site. Imposed Leaf Area Index ($LAI$) in red, measured $LAI$ in blue, vegetation height in purple and measured vegetation height in orange.

### 3.3 Dry rainfed natural grass site: Els Plans

The Els Plans site is a rainfed, relatively dry area with natural grass that was drying during the LOP. The parcel is located within a special protection area for steppe birds, and it is not cultivated. The energy budget of a short dry-down period near the end of the LOP (Fig. 3) shows a lower $Rn$ compared to La Cendrosa, because the contribution of the net long wavelength radiation is lower and the soil albedo is lower. The small amount of water available in the soil makes the $H$ the dominant term to compensate the $Rn$. The $G$ can reach values twice as high as the $LE$, except after rain events, when the evaporation from the 290 bare ground peaks. The energy budget residual at this site tends to be greater in the early morning, and it is reduced after the entry of the Marinada, a local sea breeze wind (Jiménez et al. 2023). This behavior is also observed in La Cendrosa when the vegetation is low. As a colder wind (Lunel et al. 2024), the Marinada advects moisture and cool air, and contributes negatively to the energy budget, bringing the residue closer to zero.

### 4 Model configuration

### 4.1 Forcing data

The simulations of La Cendrosa and Els Plans are performed offline (i.e. driven by observations as input). The associated atmospheric variables are the incident short and longwave radiation fluxes, wind speed, temperature, specific humidity, pressure, atmospheric $CO_2$ concentration, and rainfall rate at a 30 minute time step. All measurements were taken at 2 m except 300 the wind which was measured at 10 m and precipitation that was measured at 1 m for both stations. The time evolving vegetation properties are usually imposed using a 10-day or monthly time step: for the current study, this temporal resolution is



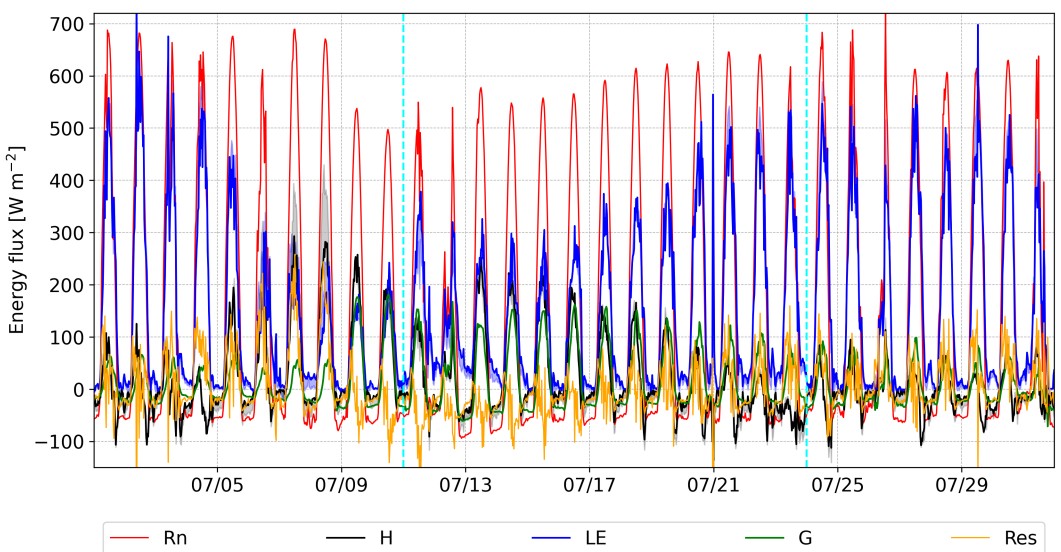

**Figure 2.** Observed terms of the Surface energy budget of La Cendrosa for the month of July 2021. In particular, $Rn$ in red, $H$ in black, $LE$ in blue, $G$ in green and the $Res$ in orange. The gray and blue shading correspond to the error in the $H$ and $LE$ due to the $Res$, which is accounted for by the Bowen ratio. Cyan dashed lines indicate irrigation events and the gray line indicates the cutting of the alfalfa.

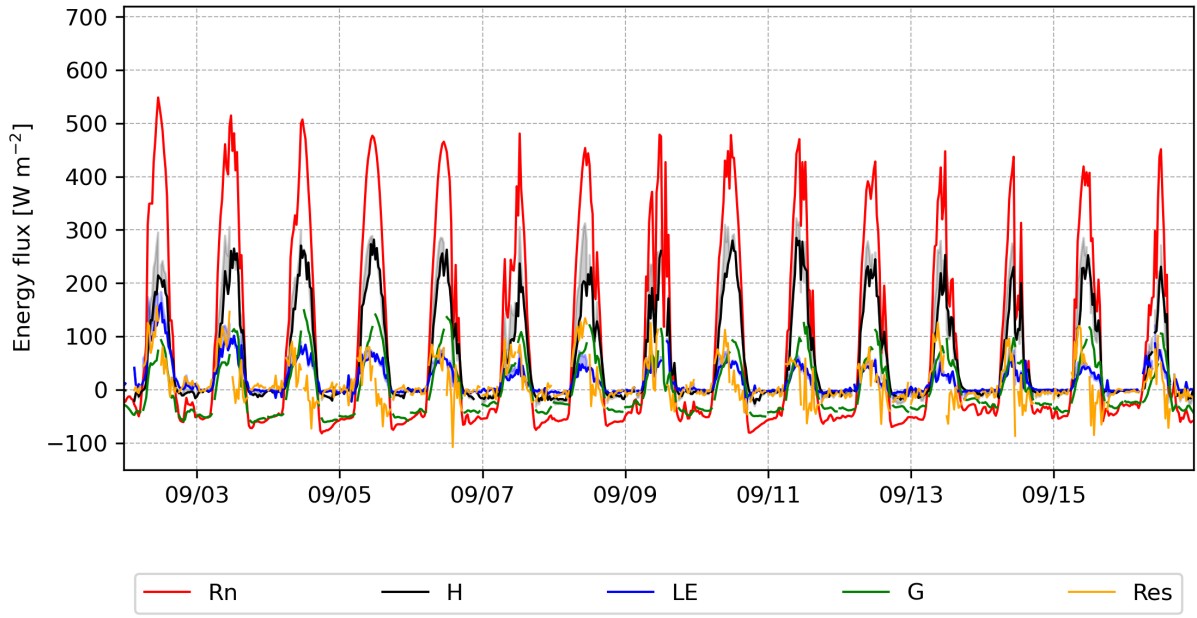

**Figure 3.** Observed terms of the Surface energy budget of Els Plans for a dry down period. In particular, $Rn$ in red, $H$ in black, $LE$ in blue, $G$ in green and the $Res$ in orange. The gray and blue shading correspond to the error in the $H$ and $LE$ due to the $Res$, which is accounted for by the Bowen ratio.




**Table 2.** Values for alfalfa of the tested variables of the AG-s configuration in SURFEX. The simulated value corresponds to the final set of parameters used in the results section.

| Variable | symbol | units | default value | simulated value | tested range | literature range |
|---|---|---|---|---|---|---|
| Mesophylic conductance | $g_m^*$ | $\text{mm s}^{-1}$ | 0.001 | 0.005 | 0.0005-0.2 | 0.002-0.01 |
| Cuticular conductance | $g_c$ | $\text{mm s}^{-1}$ | 0.00025 | 0.00025 | 0.00007-0.006 | 0.00007 |
| Quantum efficiency | $\epsilon_0$ | $\text{mg J}^{-1}$ | 0.017 | 0.0265 | 0.017-0.0265 | 0.017-0.0265 |
| Maximum assimilation | $A_{max}$ | $\text{mg m}^{-2}\text{s}^{-1}$ | 2.2 | 3.02 | 1.4-3.02 | 1.4-3.02 |

insufficient (more detail in Sect. 4.2.1). For Els Plans, several incidents in the availability of electric current after rain events resulted in gaps in the input variables. Short-wave radiation was gap-filled using a theoretical solar-zenith dependent method, and long-wave radiation gaps were filled using the correction proposed by Brutsaert (1975). The wind speed was filled with

the 3 m data at the same site, corrected by adjusting the wind speed to that at 10 m above the surface assuming a logarithmic profile at neutral conditions. Pressure was gap-filled with a fixed average value during the same period. Since the albedo of the soil changes with the variation of the water content, values every 10 days are taken into account.

For La Cendrosa, irrigation was treated as rain by adding 30 mm of water during two hours after 00 UTC. This approach introduces error for about four hours as it simulates an increase in evaporation of intercepted water over the leaves, the latent heat

increases up to 130 W m$^{-2}$ when vegetation is low and close to 70 W m$^{-2}$ when it is high. The $LAI$ and vegetation height cycle (shown in Fig. 1) cause the global albedo to change dynamically.

## 4.2 Parameter selection

Table 3 shows the parameter values for the different model configurations. The prescribed dynamic roughness length ($z_0$) for both stations falls within the lower limits of the tabulated values in the literature (Oke 2002). For La Cendrosa, a single value

is chosen as it gives reasonable estimates of the momentum for the whole period. The roughness length and the displacement height depend on the height of the vegetation but also on the density of the vegetation (Foken and Napo 2008), and the dependencies are still being parametrized (see the background in Jasinski et al. 2005) and are not currently modeled in ISBA, therefore a compensatory effect between the change on the two processes may be in play. For Els Plans, the roughness generating elements remain constant throughout the study period. The thermal roughness length ($z_{0h}$) is smaller at La Cendrosa,

which shows less steep temperature profiles near the surface in the presence of dense vegetation. The emissivity values are within the error indicated by the tabulated values and have been left equal for both sites with a value of 0.97 (Snyder et al. 1998; Simó et al. 2019). However, it should be noted that the sensitivity to this parameter has been found to be low for this parameter for the ranges encountered at the two sites studied herein. For the type of vegetation of the alfalfa field, the C3 crop type is selected and natural grass is used for Els Plans (Table 3).






**Table 3.** SURFEX parameters characterizing the simulations of the Els Plans and La Cendrosa sites. In order, the dynamic roughness length ($z_0$), heat roughness length ($z_{0h}$), the number of soil layers, the SURFEX patch identifier (7 corresponds to C3 crops and 10 corresponds to natural grass), the range of leaf area index ($LAI$) and the height of the vegetation ($Hveg$) during the study period. *The strategy has been changed to drought-tolerant.

| Site | $z_0$(m) | $z_0/z_{0h}$ | soil albedo | veg albedo | soil layers | patch | $LAI$(m$^2$/m$^2$) | $Hveg$(m) |
|---|---|---|---|---|---|---|---|---|
| La Cendrosa | 0.05 | 20 | 0.28 | 0.25 | 14 | 7* | 0.3-3 | 0.1-0.7 |
| Els Plans | 0.01 | 1 | 0.19-0.25 | 0.25 | 25 | 10 | 0.1 | 0.2 |

**Table 4.** SURFEX parameters characterizing the soil properties of the Els Plans and La Cendrosa sites. The columns, in order, represent the depth ($d$), the sand and clay content, the water field capacity ($w_{fc}$), the wilting point ($w_{wilt}$) water content, the saturated water content and soil porosity ($w_{sat}$), the saturated hydraulic conductivity ($K_{sat}$), the $b$ parameter of the CH78 pedotranfer function and the soil water potential at saturation, $\Psi_{sat}$.

| Site | $d$ (cm) | sand | clay | $w_{fc}$ | $w_{wilt}$ | $w_{sat}$ | $k_{sat}$ ($*10^{-5}$) | $b$ | $\Psi_{sat}$ |
|---|---|---|---|---|---|---|---|---|---|
| La Cendrosa | 0-10 | 38.2/52.8 | 24.4/19.95 | 0.34 | 0.10 | 0.45 | 0.452 | 6.84 | -0.33 |
| La Cendrosa | 10-30 | 56.0 | 15.5 | 0.28 | 0.10 | 0.43 | 1.488 | 5.62 | -0.23 |
| Els Plans | 0-10 | 16.3 | 35.1 | 0.38 | 0.03 | 0.48 | 0.176/0.059 | 8.31/4.89 | -0.51/-0.33 |
| Els Plans | 10-20 | 15.0 | 36.6 | 0.38 | 0.03 | 0.48 | 0.163/0.030 | 8.52/5.41 | -0.52/-0.52 |
| Els Plans | 20-30 | 35.3 | 30.6 | 0.35 | 0.10 | 0.46 | 0.283/0.177 | 7.69/5.60 | -0.35/-0.34 |
| Els Plans | 30-40 | 28.6 | 37.2 | 0.37 | 0.10 | 0.46 | 0.184/0.258 | 8.60/5.04 | -0.40/-0.18 |

### 4.2.1 Vegetation characterization of alfalfa

A realistic simulation of transpiration is the key to providing a good estimate of the $LE$, especially for the alfalfa. It is dominated by several parameters that are best prescribed using observational data (if present). The changes of $LAI$ for La Cendrosa (Fig. 1) are imposed every time step compared to the original code since the alfalfa growth is relatively rapid (see Sect. 3.2). The height of the vegetation is modeled using a linear dependence on $LAI$ based on the observations. Additionally, the AST option within SURFEX is used for both simulations. With this option, the A-gs scheme is used to model photosynthesis. The vegetation can have either drought tolerant, Eq. (10), or drought avoidant strategies, Eq. (8), (Calvet et al. 2004). The A-gs scheme is rarely compared with direct vegetation measurements (Van Diepen et al. 2022) because of the difficulty of finding a one-to-one equivalence for the parameters within the model. Direct observations of stomatal conductance ($g_s$), $CO_2$ assimilation and photosynthetically active radiation (PAR) were made at la Cendrosa. The mesophylic ($g_m^*$) and cuticular conductances ($g_c$), and the maximum assimilation ($A_{max}$) can be found from observations or via the maximum catalytic capacity of Rubisco ($V_{cmax}$), the enzyme that controls energy production. In contrast, quantum efficiency ($\epsilon_0$) is usually a fitted parameter when the energy, assimilation and carbon curves are measured. After testing multiple parameter configurations, it was found that increasing the quantum efficiency, $\epsilon_0$, and the maximum assimilation ($A_{max}$) gave results that reproduce results



obtained with a higher cuticular conductance. More energy than the default must be supplied to match the observed fluxes, regardless of whether it comes from the stomata or the cuticle. For this reason, the model adaptations are described below.

– Mesophyll conductance: The PFT parametrization establishes the $g_m^*$ through an associated curve that varies depending on whether the water use efficiency strategy under moderate stress is stress tolerant, Eq. (10), or stress avoidant, Eq. (8), and only one type is associated with each vegetation type (Calvet 2000). Although alfalfa is a C3 crop, its drought

strategy is tolerant as in C4 crops. Such plants species are known as C4-like species (Way et al. 2014). In consequence, the stress type has been changed to drought tolerant for the simulation. For this site, the highest impact comes from the increase in absolute transpiration and not from the possible changes is stomata closure. With this strategy, parameters are set for generic species and must be modified to represent a particular species. The $g_m^*$ values have been changed to $0.005\,m\,s^{-1}$ (see Table 2). These values are within the observed values of the control (Aranjuelo et al. 2013) and lower

values would correspond to stressed alfalfa. Lower values of $0.00197\,m\,s^{-1}$ were also observed with unstressed alfalfa (Malik et al. 2018). For the CLASS model (De Arellano et al. 2015), González-Armas et al. (2023) used a $g_m^*$ value of $0.01\,m\,s^{-1}$ which was found for the same site. After setting the other parameters, the differences in mean error and root mean square error modifying the $g_m^*$ value from 0.005 to 0.1 m s$^{-1}$ in SURFEX are below one W m$^{-2}$ for all fluxes. Differences arise on the days on which stomatal closure is simulated and depend on what degree the closure is effective.

It should be noted that alfalfa is bred for the development of drought/salinity tolerant varieties that are thus optimized for cultivation in arid and semi-arid regions of the Mediterranean area (Mouradi et al. 2022). However, the modification of this parameter is necessary, but there are still errors in the estimation of ET so additional parameters are explored.

– Cuticular conductance: As a first approximation, increasing the cuticular conductance to $0.0006\,m\,s^{-1}$ provided good estimates of ET. Measurements of cuticular conductance can follow several strategies and give different values for the

same species (Kerstiens 1996). Simplification to a minimum value of conductance ($gmin$) and other similar values such as nocturnal conductance or detached leaf conductance are more commonly measured (Duursma et al. 2019), $gmin$ values are more similar to values of $g_c$ which serve as a default in numerical simulations and are used as a proxy. For alfalfa, the value was set to $0.00007\,m\,s^{-1}$ according to Kerstiens (1996). When used, it provided simulations with a profound dip in evaporation in the central part of the day, resulting in unrealistic results compared to observations. No

further values of $g_c$ for alfalfa were found in the literature, which could probably indicate the difficulty of measuring this parameter. The Kerstiens (1996) value is one order of magnitude below the tested value of $0.0006\,m\,s^{-1}$ and three times less than the default value in SURFEX of $0.00025\,m\,s^{-1}$. The default value has been used as it is closer to the literature value without being outside the typical observed values of this parameter for other vegetation. Values up to at least $0.006\,m\,s^{-1}$ can still provide reasonable estimations of ET but are outside the reported values and they ar non-

physical. This behavior is given by the relationships from equations (17) and (6), which holds the sum of $g_c$ and $g_m^*$ in the denominator and can compensate the effects from one another. Together with Eq. (11), an increase of $g_c$ produces a similar effect to increasing $g_m^*$.



– Quantum efficiency: The $\epsilon_0$ can be directly related to the increase in transpiration, an increase of $\epsilon_0$ in Eq. (21) induce an increase in carbon assimilation in Eq. (20) and of the stomatal conductance, Eq. (11). This parameter is based on the theoretical quantum requirement of photons needed to assimilate one molecule of $CO_2$ (Von Caemmerer and Farquhar 1981), and it limits the assimilation rate (see review in Van Diepen et al. (2022)). Goudriaan et al. (1985) presents a revised value of $0.017\,\mathrm{mg}\,CO_2\,\mathrm{J}^{-1}$ and suggests that selective breeding of the plant species could lead to an increase in this value in new varieties. Its value has not evolved in the SURFEX configuration over the years. There is some evidence that this value is variable; Collatz et al. (1992) showed a small temperature dependence and Cai and Dang (2002) identified a variability of this parameter within tree species, but the exploration of this parameter in measurements has seemingly not progressed in recent years for meteorological applications. Mouradi et al. (2022) and Jiang et al. (2009) show that depending on the alfalfa variety, the electron transport rate and the PAR conversion efficiency (biological measurements used by their community) can be increased, apparently exploiting an increase in electron efficiency, although no direct conversion to these parameters has been found. This efficiency will decrease during drought events (Zhang et al. 2019; Mouradi et al. 2022). Thus, the $\epsilon_0$ value used may be more variable than reported and may have been compensated by an increase in other parameters such as cuticular conductance. The $\epsilon_0$ value of $0.0265\,\mathrm{mg}\,CO_2\,\mathrm{J}^{-1}$ found by González-Armas et al. (2023) increases the $LE$ to values closer to the observed values and is therefore used in this study.

– Maximum assimilation: The $A_{max}$ sensitivity is most relevant for low $LAI$ values, as marginal increases of this value will be weighted more heavily to increase transpiration than for other parameters. The relationship of $A_{m,max}$ in Eq. 15 has a nonlinear response for $g_s$. The increase of $A_{m,max}(25)$ in Eq. (19) increases $A_n$ in Eq. (20) and on to the stomatal conductance, Eq. (11), but $A_{m,max}$ can also be found in the denominator.

The default value of $A_{m,max}(25)$ is $2.2\,\mathrm{mg}$. For alfalfa, both higher values (Bunce 2018) with $2.64\,\mathrm{mg}$, which were transformed from $V_{cmax}$ as in Collatz et al. (1991), and lower values (Malik et al. 2018) of $1.4\,\mathrm{mg}$ have been reported in the literature. González-Armas et al. (2023) found values ($3.02\,\mathrm{mg}$) at La Cendrosa that were considerably higher than those given in the literature and the standard value in SURFEX. Increasing this value in this simulation improves the transpiration estimates for days that show a dip in ET during the day reducing its intensity, therefore the value in González-Armas et al. (2023) is kept, but the default values are adequate for most of the simulated days.

#### 4.2.2 Vegetation characterization of drying grass

For Els Plans, the choice of $LAI$ is complex. Measurements of $LAI$ at the site have a median value of $0.34\,\mathrm{m}^2/\mathrm{m}^2$ and a minimum value of $0.12\,\mathrm{m}^2/\mathrm{m}^2$ (Brooke 2023). However, these measurements take into account the shaded area and do not necessarily represent active vegetation. In contrast, in ISBA $LAI$ is taken as green $LAI$ and therefore represents fully active vegetation (Brut et al. 2009). This does not correspond well with the measurements in Els Plans, where the vegetation is dying but still provides shade and intercepts moisture. Furthermore, this shade comes from the blades of the natural grass rather than



the leaves. The value of $LAI = 0.1 \, \mathrm{m^2/m^2}$ is used as a compromise to take into account the shaded ground and part of the dry vegetation that does not contribute to evaporation, thus limiting transpiration.

A complementary parameter necessary to characterize the vegetation is vegetation height ($Hveg$), which tests show is not a highly sensitive parameter for either site and is set to 0.2 m by rounding the observed value of 0.16 m for Els Plans. In healthy active low vegetation such as crops, $LAI$ may not be directly related to its height or it may depend on the crop so they are set

independently (Yuan et al. 2013).

### 4.2.3 Soil characterization

Table 4 lists the measured sand and clay content values together with the soil hydraulic parameters obtained from the soil texture data and the pedotransfer function from Clapp and Hornberger (1978), referred to herein as CH78. For la Cendrosa, two surface measurements of the soil characteristics were carried out by different teams. The first was near the position of

the SEB station with two levels, the second was about 20 m away within the same field. It was observed that the second measurement had proportions of sand and clay close to those at a depth of 10-30 cm of the first measurement. Since the field is irrigated by flooding, some washing of the soil can take place, which can be the source of these near-surface discrepancies in texture between the samples of the different teams. The two-level measurements are used for La Cendrosa for all parameters except wilting point of the model, which is reduced to include the minimum water content within the root zone which is seen

in the observations. At the Els Plans site, a more extensive sampling of the soil was performed and incorporated, so that the matric potential at saturation $\Psi_{sat}$, the slope of the retention curve $b$ and the saturated hydraulic conductivity $K_{sat}$ values were fitted from soil core samples using the HYPROP method (Shokrana and Ghane 2020) and $Ksat$ values were measured in the laboratory. The $\Psi_{sat}$ value differed from that given by CH78 at the surface and bottom of the soil profile and the $b$ parameter is at the lower end of its range. Using these vertically-varying observed values reduced the overall error of the simulation

compared to using a constant values for the vertical profile of the soil and compared to the default values taken from the global database. This matches the results of Sobaga et al. (2023) who identified that the default $b$ parameter produced a lack of drainage in SURFEX V8.1. $K_{sat}$ is taken as the observed value instead of the fitted one, as it also leads to an improvement in the $LE$ and $H$ estimation. We note that it differs by an order of magnitude at the surface. Soil layers deeper than those observed were assumed to have the same soil properties as that in the lowest observed layer. The discretization of the soil layers has been

chosen to match the layers to the depth of the observations, while maintaining the highest vertical resolution near the surface of the soil column.

## 5 Results/Discussion

This section presents an analysis of the simulations carried out for La Cendrosa and Els Plans using the default and new soil resistance parametrizations. The different simulations are identified by the names NON, S92 and DSL to indicate no soil

resistance, the use of Sellers 92 resistance and a DSL resistance, respectively.





According to Foken and Napo (2008), a bulk indication of instrumental errors indicates that $Rn$ can have errors of around 10 %, errors in $H$ and $LE$ can reach 20 %, together with a 50 % error for the $G$, not accounting for the energy storage error. Note that budget closure is not necessarily a measure of the quality of the fluxes (Aubinet et al. 2000). The lack of closure in observations is expressed as residual energy. It includes part of the instrumental error, but also owing to horizontal advection
due to heterogeneity (not modeled), heat storage due to vegetation and its exchanges (modeled in ISBA-MEB but considered small compared to soil storage under normal circumstances), and unmeasured phase water changes in the soil (not modeled) (Cuxart et al. 2015).

As mentioned in Sect 3.3, the local sea breeze reduces the residue, so some advection contribution to the residue is expected. Consequently, the main analysis concentrates on the comparison of fluxes with the simulation without imposing closure on the
observed fluxes but the statistical comparison is also shown with the closure imposed by the Bowen ratio method (Barr et al. 1994).

### 5.1   The simulated energy budget

Table 5 shows the statistical comparison of the energy budget for La Cendrosa, divided into three time periods; daily, daytime and night. The distinction between daytime and night is made from 8 a.m. to 8 p.m., leaving out the more complex changes of
the transitions in the night period since the main changes caused by the use of a resistance occur during the day. The following analysis focuses on the daytime, so that behaviors resulting from certain periods, such as low and high vegetation or flooding events, can be identified. A comparison of the simulated and observed $Rn$ is shown in Fig. 4 a and e. Despite the mean error (ME) indicating a bias close to zero, the NON simulation is positively biased overall, and this behavior is improved through the use of the DSL scheme. In particular, two days show a different trend that modifies the statistics (Table 5). This behavior
corresponds to the period between the cutting and the removal of the alfalfa, which took place a few days later. The effect that the resistances have on the $Rn$ is mainly due to the difference in the longwave energy budget and surface temperature, since less heat is emitted in the NON simulation by the soil, and the albedo input does not change between simulations. The $LE$ flux shown in Fig.s 4 b and f is centered on the one-to-one line, with a subset that is deviating from the one-to-one line corresponding to the period of low vegetation with relatively low $LAI$ after the irrigation period. The improvement in $LE$ is
shown by a reduction in ME, root mean square error (RMSE) and an increase in correlation (see Table 5). For the $H$, the larger errors are corrected with the use of a resistance to compensate for the change in $LE$. The improvement of the statistics is also present for $H$ and the underestimation is reduced within the values of the standard deviation observed for the residue for La Cendrosa (56 W m$^{-2}$) and Els Plans (50 W m$^{-2}$). On the other hand, the simulated soil stores more energy and releases it faster than observed, with a daily cycle characterized by a hysteresis effect that overestimates during the day and underestimates at
night thus worsening the results for $G$. The resistance also impacts the nighttime statistics with a reduction of the $Rn$ that results in a lower available energy at night, and an small increase of ME and RMSE for $LE$ and $H$ of the order of 1 W m$^2$.

At the Els Plans site, the $Rn$ is well represented by the imposed albedo (Fig. 5). The error of $Rn$ (Table 6) remains well below the 10 % error of the measurement. The use of a DSL resistance increases the RMSE and ME up to 2 $W\ m^{-2}$. For the $LE$, the scatter is large in all three simulations compared to its absolute value, with a delay in the start of evaporation





and an underestimation with absolute errors close to 50%. The RMSE of $LE$ is reduced during the daytime from 29 W m$^{-2}$ to 19 W m$^{-2}$ using the DSL approach. The correlation is best for the S92, followed by the NON simulation, and closely by the DSL simulation as in this case the $LE$ is too damped. The sensitivity to the parameters used is discussed in Sect. 5.4. For the $H$ there is a small reduction in the RMSE but an increase in the ME, so the overall performance remains the same except for an improvement from 0.81 to 0.88 in the overall correlation. For the $G$ performance, the intense heating of the

soil is again simulated with hysteresis, with the transfer of energy being quicker in the model than that observed (Fig. 5d and h), which corresponds to a daily RMSE of 57 W m$^{-2}$ (Table 6). No measurements of the thermal properties of the soil have been made, and so the default properties assigned to the observed soil texture have been taken. A sensitivity test indicated that reducing the thermal conductivity of solids by 25%, value within the range of different soil types reviewed from observations by Zhang and Wang (2017), could reduce this error by reducing the $G$ flux up to 20 W m$^{-2}$, while increasing the error in the

$H$ proportionally. Overall, due to the higher error in the measurements of $G$, estimated at 50%, this bias in the simulation is considered tolerable. The change of $Rn$ and $H$ at night due to the resistances is the same as for La Cendrosa, but in this case, $LE$ has an improvement of the correlation of approximately 0.08, with a nearly negligible change in the overall average $LE$ value.

## 5.2   The latent heat flux

The observed $LE$ is shown for the aforementioned simulations in Fig. 6 a together with a top to bottom barplot with rain and irrigation. The largest change between simulations is after the first irrigation event. The increase of observed $LE$ is overestimated with the NON option, mainly due to the large contribution from soil evaporation, as in (Lohou et al. 2014), whereas the $LE$ in the DSL simulation is reduced more rapidly, which matches more accurately the observations in this period. The other irrigation event and small rain events result in small increases in $LE$ that decreases during the hours after as the soil

is still near saturation and have little impact on the available water.

Estimates during periods of high $LE$ are well captured by all three simulations, and it is slightly better for the NON simulation, than for the S92 and DSL simulations since the reduction of soil evaporation is not compensated by increased transpiration. The differences range from 5 to 10% as the soil dries. $LE$ is also well represented for the low vegetation period for the three simulation types. The soil resistance reduces the contribution of soil evaporation to $LE$ resulting in an improvement in the

RMSE of $LE$ (daily) of 16 W m$^{-2}$ for the S92 simulation and 20 W m$^{-2}$ for the DSL simulation (see Table 5).

The nocturnal $LE$ values show good agreement with the observations, except for nights with a higher flux, in which the simulations underestimate the value (e.g. July 8th). The simulations respond well to small rain events such as the night of the 21st but with a more moderate increase. Such an increase occurs artificially in the simulation for the first irrigation event reaching 130 W m$^2$, as irrigation was imposed as rain, but returns to the observed values after five hours.

For Els Plans, the interruption of the measurements after rain events mentioned in Sect. 4.1 affected the number of dry-down periods that were well captured, limiting the analysis of such events. Such a period is shown where the maximum $LE$ is better captured by S92 in Fig. 6 b compared to the NON or DSL. Although this is after a rain event, the $H$ is dominant and the magnitude of $LE$ is small. The vegetation at this site is mostly senescent, and the water content of the first soil layers is very



**Table 5.** Mean Error (ME) and Root Mean Square Error (RMSE) in W m$^{-2}$ for the net radiation ($Rn$), the sensible heat flux ($H$), the latent heat flux ($LE$), and the ground heat flux ($G$) for La Cendrosa site. In brackets is the modified value taking into account the residual using the Bowen ratio method.

| daily | ME $Rn$ | RMSE $Rn$ | ME $H$ | RMSE $H$ | Corr | ME $LE$ | RMSE $LE$ | Corr | ME $G$ | RMSE $G$ |
|---|---|---|---|---|---|---|---|---|---|---|
| NON | 0.67 | 31.65 | −4.44 (−18.06) | 45.90 (55.19) | 0.81 (0.84) | 16.51 (−10.43) | 65.85 (70.70) | 0.93 (0.92) | −10.80 | 56.68 |
| S92 | −2.46 | 29.21 | −0.49 (−14.11) | 39.09 (47.70) | 0.88 (0.91) | 8.25 (−18.68) | 50.03 (62.58) | 0.95 (0.94) | −9.62 | 60.99 |
| DSL | −5.49 | 28.00 | 4.52 (−9.10) | 37.11 (43.86) | 0.88 (0.90) | −0.68 (−27.61) | 46.70 (65.04) | 0.95 (0.94) | −8.73 | 64.90 |
| daytime | ME $Rn$ | RMSE $Rn$ | ME $H$ | RMSE $H$ | Corr | ME $LE$ | RMSE $LE$ | Corr | ME $G$ | RMSE $G$ |
| NON | 7.35 | 41.09 | −18.85 (−33.83) | 59.75 (73.76) | 0.80 (0.83) | 35.49 (−1.84) | 87.22 (86.64) | 0.87 (0.86) | −1.69 | 64.88 |
| S92 | 3.01 | 37.32 | −12.03 (−27.01) | 49.55 (63.01) | 0.88 (0.90) | 19.10 (−18.23) | 64.28 (73.96) | 0.92 (0.90) | 3.55 | 70.28 |
| DSL | −0.76 | 35.02 | −4.47 (−19.45) | 45.70 (57.20) | 0.87 (0.89) | 3.04 (−34.29) | 58.64 (77.27) | 0.92 (0.91) | 8.28 | 75.02 |
| night | ME $Rn$ | RMSE $Rn$ | ME $H$ | RMSE $H$ | Corr | ME $LE$ | RMSE $LE$ | Corr | ME $G$ | RMSE $G$ |
| NON | −6.68 | 17.25 | 10.18 (−2.17) | 24.65 (23.81) | 0.83 (0.86) | −4.35 (−21.04) | 28.93 (49.20) | 0.89 (0.84) | −18.49 | 48.60 |
| S92 | −8.55 | 17.52 | 11.08 (−1.27) | 24.47 (23.25) | 0.85 (0.87) | −4.12 (−20.80) | 27.55 (48.54) | 0.90 (0.85) | −21.49 | 51.75 |
| DSL | −10.78 | 18.53 | 13.39 (1.04) | 26.15 (23.74) | 0.82 (0.84) | −5.56 (−22.25) | 29.10 (49.87) | 0.89 (0.84) | −24.59 | 54.43 |

**Table 6.** Mean Error (ME) and Root Mean Square Error (RMSE) in W m$^{-2}$ for the net radiation ($Rn$), the sensible heat flux ($H$), the latent heat flux ($LE$), and the ground heat flux ($G$) for Els Plans site. In brackets is the modified value taking into account the residual using the Bowen ratio method.

| daily | ME $Rn$ | RMSE $Rn$ | ME $H$ | RMSE $H$ | Corr | ME $LE$ | RMSE $LE$ | Corr | ME $G$ | RMSE $G$ |
|---|---|---|---|---|---|---|---|---|---|---|
| NON | −4.33 | 11.39 | 12.06 (−15.54) | 45.37 (62.00) | 0.92 (0.89) | 2.80 (−5.04) | 22.13 (23.92) | 0.77 (0.75) | −7.35 | 70.05 |
| S92 | −4.75 | 11.79 | 12.69 (−14.91) | 44.00 (60.66) | 0.92 (0.90) | 1.83 (−6.00) | 15.71 (21.41) | 0.81 (0.75) | −7.32 | 73.39 |
| DSL | −5.17 | 12.23 | 13.72 (−13.88) | 44.50 (60.88) | 0.92 (0.90) | 0.34 (−7.50) | 15.37 (22.75) | 0.78 (0.72) | −7.36 | 76.28 |
| daytime | ME $Rn$ | RMSE $Rn$ | ME $H$ | RMSE $H$ | Corr | ME $LE$ | RMSE $LE$ | Corr | ME $G$ | RMSE $G$ |
| NON | −2.76 | 14.04 | 16.12 (−22.08) | 57.91 (76.81) | 0.87 (0.84) | 5.88 (−3.73) | 28.54 (28.80) | 0.74 (0.73) | −7.48 | 81.61 |
| S92 | −3.02 | 14.43 | 16.52 (−21.68) | 55.66 (74.71) | 0.89 (0.86) | 4.35 (−5.26) | 19.58 (24.96) | 0.77 (0.71) | −6.53 | 85.97 |
| DSL | −3.29 | 14.83 | 17.53 (−20.67) | 55.98 (74.79) | 0.89 (0.86) | 1.64 (−7.97) | 19.02 (26.94) | 0.72 (0.65) | −5.31 | 89.71 |
| night | ME $Rn$ | RMSE $Rn$ | ME $H$ | RMSE $H$ | Corr | ME $LE$ | RMSE $LE$ | Corr | ME $G$ | RMSE $G$ |
| NON | −6.18 | 7.75 | 6.49 (−10.79) | 26.99 (44.09) | 0.89 (0.88) | −0.71 (−6.83) | 12.75 (17.98) | 0.61 (0.58) | −4.03 | 58.71 |
| S92 | −6.77 | 8.24 | 7.41 (−9.88) | 27.39 (44.07) | 0.89 (0.88) | −1.35 (−7.46) | 10.40 (17.58) | 0.69 (0.66) | −4.73 | 60.92 |
| DSL | −7.32 | 8.76 | 8.42 (−8.87) | 28.47 (44.65) | 0.88 (0.88) | −1.58 (−7.70) | 10.64 (18.14) | 0.72 (0.68) | −6.01 | 62.83 |





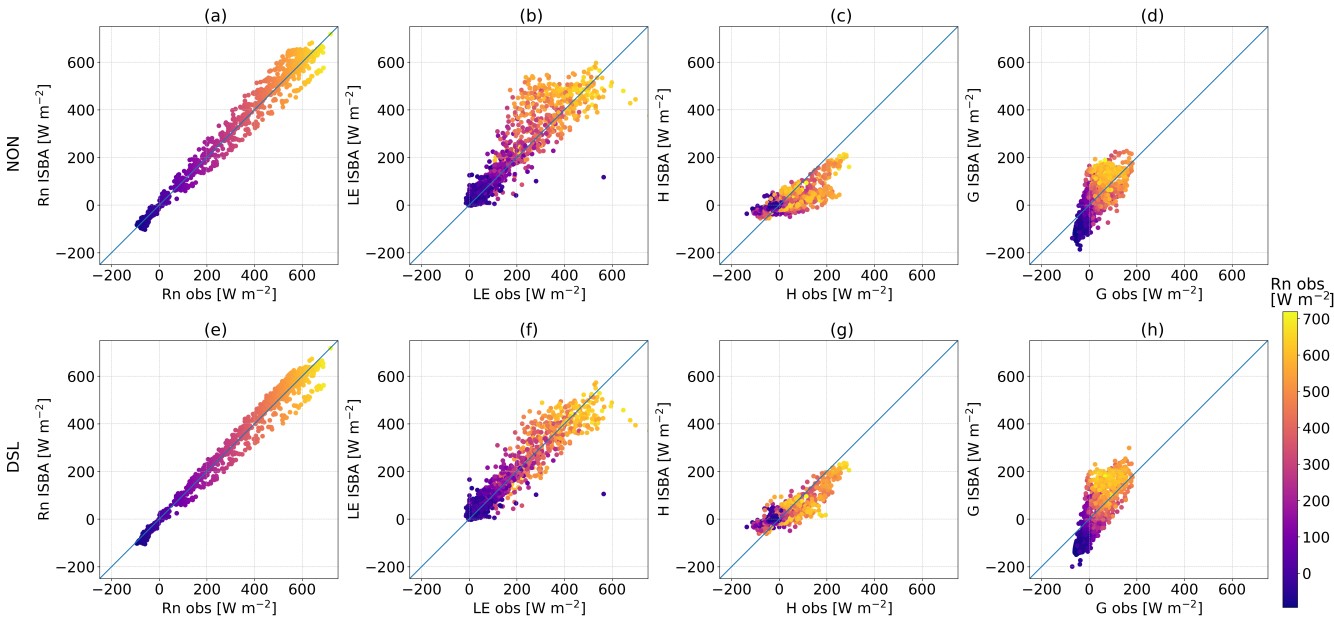

**Figure 4.** Scatterplots of the simulated terms of the energy budget against the observation for La Cendrosa site for the NON simulation (NON, a-d) and the DSL simulation (DSL,e-h). From left to right, $Rn$ (a,e), $H$ (b,f), $LE$ (c,g), and $G$ (d,h).

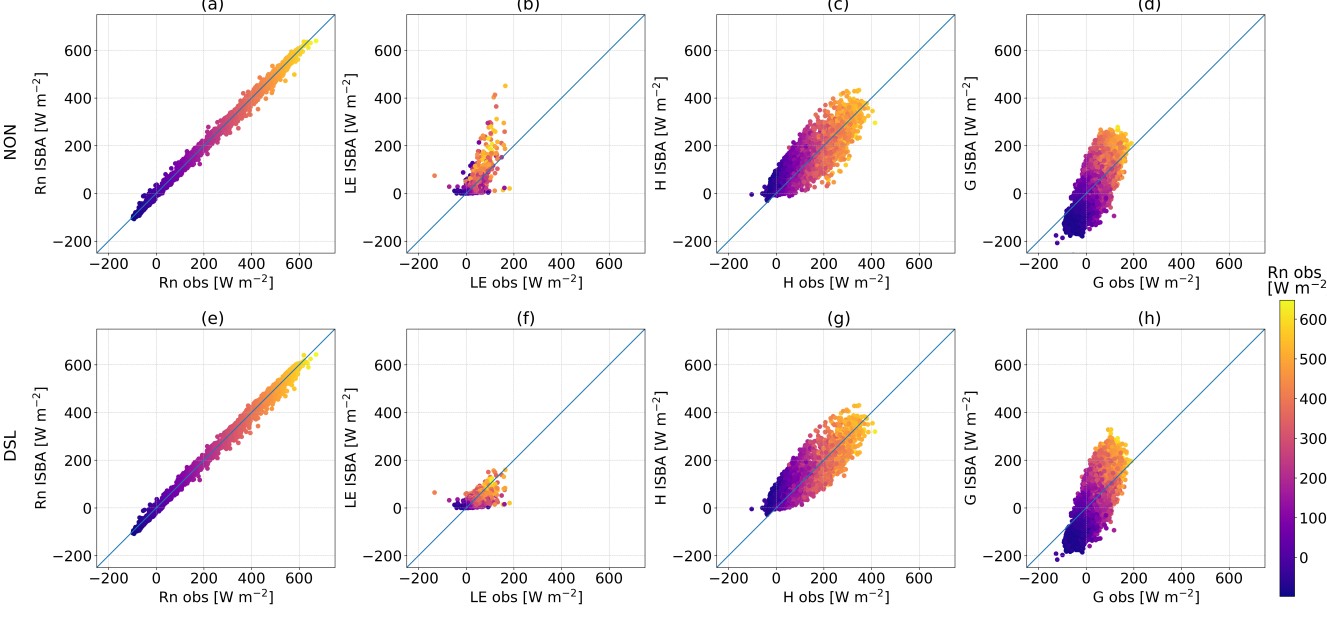

**Figure 5.** Scatterplots of the simulated terms of the energy budget against the observation for Els Plans site for the simulation with no resistance (NON, a-d) and simulation with the DSL approach (DSL,e-h). From left to right, $Rn$ (a,e), $H$ (b,f), $LE$ (c,g), and $G$ (d,h).





low. Following a rain event, soil evaporation increases and contributes significantly to ET throughout the day. Evaporation from
the interception is also present in the model for about six hours and increases up to $10\,\mathrm{W\,m^{-2}}$. However, as the soil dries, bare
soil evaporation contribution is present only in the middle hours of the day. Instead, the main contribution is attributed to the
dying vegetation and shows a slower daily cycle with a slower progression in the increase in ET. The modeled transpiration
values are on the order of $30\,\mathrm{W\,m^{-2}}$ compared to the daily maximum of $LE$ of $50\,\mathrm{W\,m^{-2}}$. Nonetheless, they represent only
a relatively small part of the energy budget and are of the order of the residue. For Els Plans the nocturnal observations show
that negative $LE$ values are common at night, indicating dew formation or soil water vapor adsorption, which is typical not
only under such dry conditions (Paulus et al. 2024; Kohfahl et al. 2021), but also under more humid soil conditions (Groh et al.
2018a). The presented simulations have struggled to reproduce these negative values, because the soil temperatures are high
and the transition to a stable regime is rare.

### 5.2.1 Soil resistances

Modeled soil resistances are at their minimum values after rain or irrigation events and increase with time as the soil dries.
For La Cendrosa site (Fig. 7a), the soil resistance value increases using the DSL approach and resistance values can reach up
to four times that of S92. Note that the DSL resistance does not start acting until the $VWC$ has fallen below saturation to the
$w_{dsl0}$ threshold, unlike the S92, which always presents some resistance.

In the case of Els Plans, the resistance values of the DSL simulation remained continuously above the value of S92, and
increased by a factor of almost 2 (Fig. 7b). Generally, rainfall events were not abundant enough during the summer season with
high evaporative demand to reduce the resistances to zero at this site. Although they were significantly reduced at times, the
arid conditions were persistent throughout the summer months.

The resistance values estimated for Els Plans are similar to those found in Swenson and Lawrence (2014), while those of La
Cendrosa are higher, due to the difference in soil properties. The increase in resistance starts earlier than observed in laboratory
studies (Zhang et al. 2015). Their values were closer to the S92 simulation, but slightly higher and remained lower than those
shown for the DSL simulation. In addition, Balugani et al. (2023) found that a DSL observed under natural conditions can be
larger than that measured in a lysimeter, whether in laboratory or field conditions. The higher values for resistance compared
to the ones by Zhang et al. (2015) may be explained by the exposition to the atmospheric conditions which will affect ET, soil
moisture and soil temperature profile (Balugani et al. 2023). To explore this further, a sensitivity analysis is carried out in the
following section to test the optimal parameter configuration.

### 5.2.2 Water Storage

To study the longer term effects of using a soil resistance, Fig. 8a shows the cumulative water loss and its decomposition into
transpiration and soil evaporation at La Cendrosa. The differences in cummulative $LE$ loss between the various simulations
are small before the vegetation is cut, up to $3\,\mathrm{mm}$ (2%). The $LE$ is underestimated in all three simulations and the simulated
transpiration dominates until the alfalfa harvest on July 6. Thereafter, a first trend change is observed, which is reproduced by
all three simulations. After the irrigation event of the 11th of July, a second change in tendency occurs for which transpiration





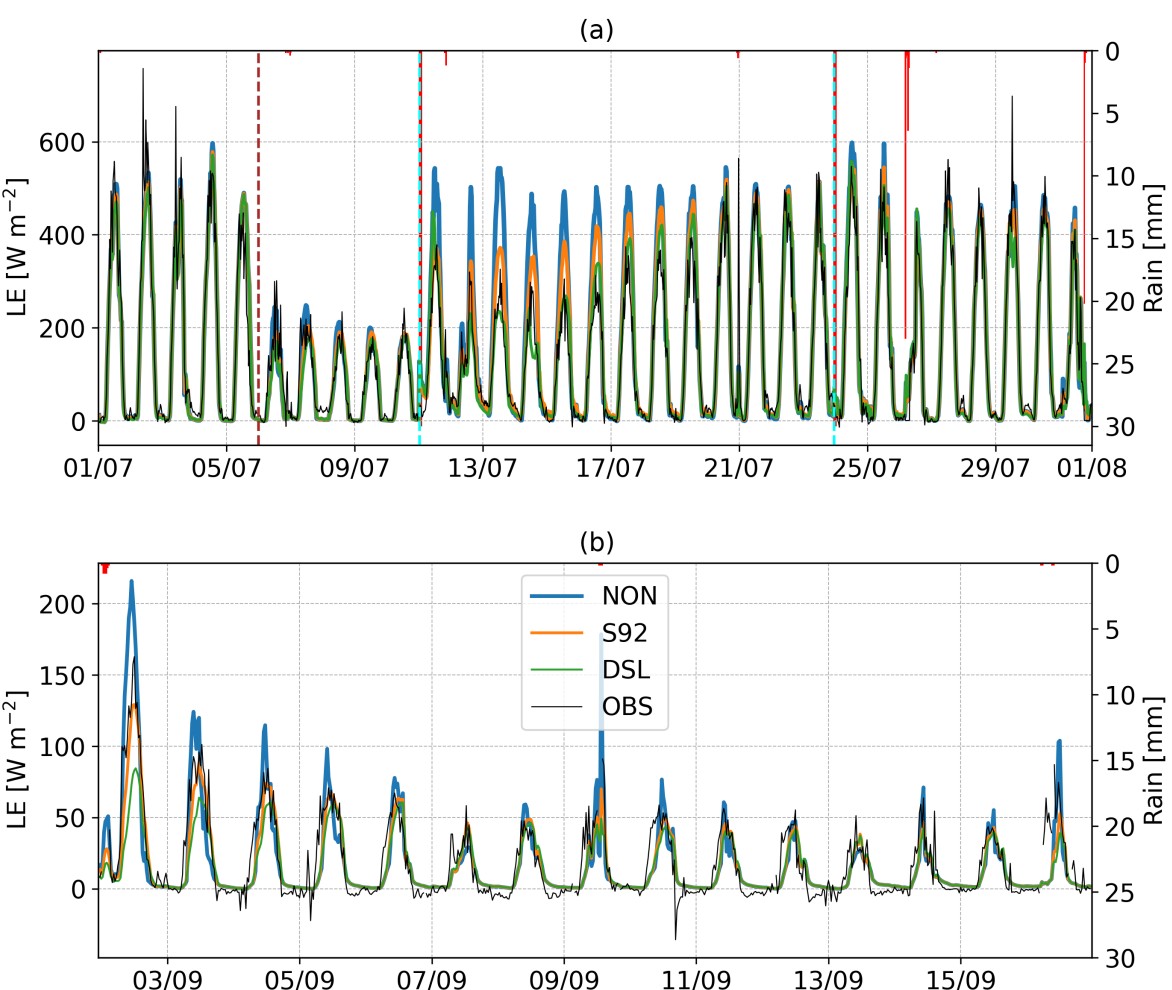

**Figure 6.** Observed (black) and simulated latent heat flux ($LE$) timeseries of La Cendrosa (a) and for Els Plans sites (b) for the simulations with no resistance (NON, blue), with a Sellers 92 resistance (S92, orange) and a DSL resistance (DSL, green). Bar lines represent the rain. Vertical lines represent the vegetation cut (brown) and irrigation (cyan).



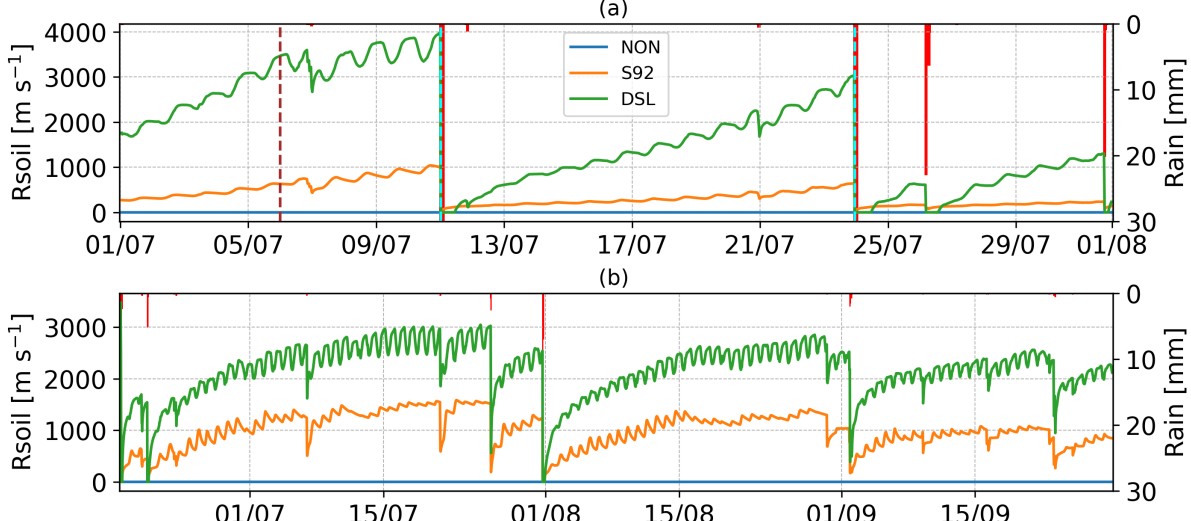

**Figure 7.** Simulated soil resistance timeseries of La Cendrosa (a) and for Els Plans sites (b) for the simulations with no resistance (NON, blue), with a Sellers 92 resistance (S92, orange) and with a DSL resistance (DSL, green). Bar lines represent the rain. Vertical lines represent the vegetation cut (brown) and irrigation (cyan).

increases slightly, but soil evaporation increases significantly the following days, the strength of which is determined by the resistance applied to the simulation (Fig. 7). The $LE$ in the DSL simulation initially underestimates ET slightly, but recovers during the growth period and matches the $LE$ trend at the end of the period, falling within 1 mm difference with respect to observations. The other two simulations accumulate an excess water loss of 9 mm for S92 and 19 mm for NON due to the overestimation of bare soil evaporation. Estimation of the partitioning can be made for July 29th at this site as an estimate of the ratio between transpiration and ET for daytime was made on July 29th using microlysimeter and EC measurements. The observation gives a ratio of 0.87 for this day. The simulations for DSL gave ratios closest to the observations with a ratio of 0.88. The NON simulations resulted in a much lower ratio (0.66), showing that the contribution of soil evaporation to total ET is significantly overestimated without DSL. However, it should be noted that the NON simulations can reach values up to 0.85 for other days.

For the four months analyzed at Els Plans, the minimum ET change was determined to be in July. After a significant rain event in early August, the ET rate increased as more water was available. Fig. 8b shows the cumulative ET and the partitioning of the model. It is important to note that we are at the limits of the model for representing ET with the current $LAI$ formulation of SURFEX. $LAI$ is used to represent the density of photosynthetically active vegetation. For this site, there was a shading effect and the thermal inertia due to the vegetation mass was high, but a very limited transpiration. These effects are not well taken into account due to their dependence on $LAI$ and its low value to represent the small part of active vegetation, and thus the soil flux is also accentuated for this site by the model.





In spite of these difficulties, the simulation with a DSL manages to reproduce the ET at the beginning of the period and mostly matches the tendency during the summer months. The split between transpiration and bare soil evaporation may be biased towards more transpiration than occurred in reality as estimations of ET in similar conditions can be of only 10% (Ma et al. 2020), but they are within the realm of possibility as senescent leaves have been reported to be limited to transpiration of 0.1 mm h$^{-1}$ (Pérez-Anta et al. 2024). The presence of gaps in the measurements of ET within the rainy period did not allow the observation of the increase of the cumulative evaporated water during periods with infiltration. The cumulative water during the driest period near the 15th of July was underestimated by 4 mm and by 10 mm for the DSL and NON simulations, respectively, during the SOP. At the end of September, the cumulative ET overestimation is slightly reduced to 1 mm and 9 mm for the DSL and NON simulations, respectively. A small amount of resilient vegetation was observed to germinate at the end of August, which may explain the additional $LE$ and reduction of the model bias. The ET values for this period are low enough that the observational uncertainty of ET can explain a large part of these differences. Nevertheless, it is noteworthy that the change in the trend of evaporation in August in the observations corresponds to the beginning of the period of germination of sporadic natural grasses at the Els Plans site.

When comparing the sites, the difference in the amount of water evaporated between the two sites is remarkable. In one month at La Cendrosa, the amount of water evaporated is three times greater than that accumulated in more than three months at Els Plans: irrigation significantly alters the water balance of the area.

## 5.3 Other relevant variables

### 5.3.1 Albedo

The observed daily variability of albedo and its model counterpart are shown for La Cendrosa in Fig. 9a. The observed albedo values varied by more than 0.1 during the month of July. The alfalfa albedo variability has been reported to be up to 0.2, depending on the solar angle (Al-Yemeni and Grace 1995). The change is linked to alfalfa leaves which track the solar angle: they cup and reduce the albedo (Travis and Reed 1983). This process occurs in conjunction with a midday decrease in leaf water potential (Bell et al. 2007).

Model albedo is obtained through a constant imposed value for both surface albedo and vegetation albedo. It was chosen to provide a more accurate $R_n$ so that the available energy of the model remained accurate for the other fluxes. ISBA-MEB modifies this initial value taking into account $LAI$ and solar zenith angle. The model reproduced then part of the cycle, but its amplitude was weaker than the measured one, changing by at most 0.03. Observations show that alfalfa has albedo values close to 0.16 when fully grown, while after cutting the surface albedo increases to values close to 0.22, varying with vegetation growth and $VWC$. As albedo was forced with daily varying values in the current study, modelling the absorption in MEB through the canopy captures some of the changes due to growth from the 9th to the second irrigation period, the 24th. In Fig. 9b, the albedo of Els Plans is influenced by the dryness of the soil, varying with the formation of the DSL and the reduction in $VWC$ in accordance to what De Silans et al. (1989) and Van de Griend and Owe (1994) have observed under bare soil semiarid conditions. The change in albedo over the LOP ranged from 0.18 to 0.25, which is consistent with the findings of







**Figure 8.** Accumulated latent heat flux ($LE$) for La Cendrosa (a) and Els Plans sites (b). Observations are in black, simulations with no resistance are in purple, with Sellers 92 in deep blue and a DSL resistance in cyan. Solid lines correspond to the $LE$, dashed lines correspond to the ground evaporation and dashed and dotted lines correspond to the transpiration contribution. The vertical brown line corresponds to the harvest, and cyan lines correspond to irrigation events.





**Figure 9.** Observed (blue) and simulated (red) albedo for La Cendrosa (a) and Els Plans sites (b).





Béziat et al. (2013), who found an albedo of 0.25 for senescent crops which are expected to be close to senescent natural grass. The observed rain events reduced soil albedo by up to 0.03, between before and after the events. The decadal prescription of the albedo is not enough to capture this variation. Due to the low $LAI$, no daily cycle is included in the simulation of Els Plans

although a sub-daily variability of albedo is observed. These differences do not generate a large discrepancy in the $Rn$.

### 5.3.2   Volumetric water content

The $VWC$ dynamics are highly tied to LE, and ISBA attempts to represent this linkage through several mechanisms. The amount of transpiration is limited by $D_{max}$, Eq. (7), through the water stress factor $f_2$ in Eq.(9), and bare soil evaporation by $VWC$, Eq. (23). The addition of a soil resistance, such as the DSL resistance, adds an additional dependence of $LE$ on the

$VWC$, that limits further $ET$ when the $VWC$ is scarce.

Reproduction of $VWC$ dynamics is highly dependent on soil hydraulic properties obtained for the site La Cendrosa from soil texture data and a pedotransfer function. The observations of $VWC$ at La Cendrosa (Fig. 10a) present a soil moisture profile that does not decrease with depth, but rather has a layer at 10 cm that reacts very quickly to precipitation or irrigation events. This shows that drainage after precipitation or irrigation is significantly faster than for observations at 5 cm or 30 cm.

This layer has a much higher sand content which, combined with a lower field capacity and a higher $Ksat$, results in faster infiltration of water into deeper soil layers. This layer seems to coincide with the point at which the number of roots begins to decrease, as seen when the sensors were installed. This could mean that in drier conditions in the topsoil (-5 cm), root water uptake can decrease rapidly. Soil samples were taken at depths of 0 to 10 cm and 10 to 30 cm and the analysis data (Table 3) show that there is a strong change in sand content and $Ksat$ at 10 cm depth. The large difference in the observed $VWC$ and

drainage dynamics after the events suggests that the probe represents the depth between 10 and 30 cm. The fast response of the $VWC$ during rainfall and irrigation events might indicate that water was transferred along spatially distinct pathways (preferential flow) in the soil subsurface (Guo and Lin 2018).

The $VWC$ in ISBA is simulated using the mixed-form of the Richardson equation, and thus it can model heterogeneous vertical texture or soil property profiles. In the current study, it was found that changing the soil hydraulic parameters repre-

senting the uppermost five layers (0-13.5 cm) to have a higher sand content at the first two layers and lower for the other three significantly impacts the soil properties, and the observed profile with a more humid 10 cm level can be better reproduced (not shown). Current soil world databases predict the superficial layer up to 5 cm to a resolution of 250 m (Hengl et al. 2017) and are used by multiple global models (Vereecken et al. 2019). But as seen by the difference in sand content of the two surface soil surface layer measurements from Table 4, the spatial variability of the soil is high at this site. Measurements within the

field close to the EC-tower showed for the top layer (0-10 cm) a much higher sand content (56%) compared to the observations at another location in the same field (38%) from Table 4. In consequence, the parameters have been left as those indicated in Table 4. The simulation has a positive bias of 5% during flood events caused by a lack of drainage, but the model is able to capture the tendency of the $VWC$ for the DSL simulation. Note that for the NON simulation, this tendency is underestimated. After the 24th of July, when the vegetation is almost fully developed, the differences between these simulations remain small,

but the differences against observation remain large.





For Els Plans (Fig. 10b), the same bias is observed in terms of the trend. The difference between the simulations becomes increasingly larger for the each soil water content level, which are reduced after rain events. The re-wetting of the deeper layer (i.e., 30 cm) only occurs for the largest rainfall event on the 31st of July as a delayed response, since the top soil layer was very dry and water was mainly used to refill the soil water storage and thus could not infiltrate deeper than 10 cm for the other events. The simulations showed a delayed response compared to observation when water infiltrated to the 30 cm layer. Despite the more extensive measurements of soil hydraulic parameters, some potentially important processes, such as flows through cracks or macropores (preferential flow paths) are currently not included in the model structure. In general, the use of pedotransfer functions or laboratory measurements may not be suitable for determining soil hydraulic properties for a specific site. Weihermüller et al. (2021) showed that the choice of pedotransfer function can have a large impact on soil water dynamics, so using an ensemble mean instead of a specific pedotransfer function can be a good solution to reduce this uncertainty (e.g. Krevh et al. 2023). A better method is the inverse estimation of soil hydraulic parameters based on observations of soil water dynamics for the different layers, which significantly reduces this parameter uncertainty. However, this requires sufficient in situ observations of $VWC$ and matric potential (Groh et al. 2018b; Schübl et al. 2023), which determine the field water retention characteristic, as well as sufficiently long time series that include drying and re-wetting phases. A study of this kind was carried out with SURFEX v8.1 by Sobaga et al. (2023), in which drainage was assessed and improved. Its functions are still being migrated to V9, so the observational derived parameters were given priority in this study.

### 5.3.3 Soil temperature

The diurnal cycle of soil temperature measured at 5 cm in La Cendrosa (Fig. 11a) has an amplitude of about 5°C on most days. Before irrigation, when the vegetation is very low, it reaches a maximum of about 10°C. In contrast, the simulations have a larger amplitude at the beginning. After irrigation, the NON simulation better reproduces the soil temperature diurnal pattern, the DSL simulation increases the temperature at 5 cm up to 5°C in response to the increase in $G$ due to the decrease in $LE$. For Els Plans, Fig. 11b), shows a period of drying. At the beginning, when water is more available, the differences between the NON and the DSL simulations was up to 2°C, leveling off when soil moisture at the surface layer reduces over time. The layers below 5 cm show the greatest differences, while the temperature at 10 cm maintains a maximum difference of 5°C throughout the period. These differences indicate that the thermal conductivity is too high. Attempts to reduce the amplitude of soil temperature, such as the inclusion of the impact of soil organic matter on the soil thermal properties, reduce the bias but do not completely correct it and do not correspond to a realistic characterization of the soil at the site. In order to remain as close as possible to the real behavior of the global application of the model, no change has been made to the thermal conductivity. Strategies such as a variable thermal conductivity of the first layer dependent on its thickness such as for the CLM model (Swenson and Lawrence 2014) could be adopted if a pattern is identified for larger domains.

### 5.4 Parameter sensitivity analysis

The parametrization of a DSL resistance depends on parameters that have uncertain values, as the formation of a DSL depends on the type of soil, but also on other soil properties that are often unknown. The parameters $K_{dsl}$ and $z_{dsl}$ are assumed to be the



**Figure 10.** Volumetric water content ($VWC$) of the soil at La Cendrosa site (a) and Els Plans (b) sites at different depths. Levels comprise 2, 5, 10 and 30 cm depending on the availability for the site. Observed VWC is shown in solid lines, the NON simulation in dashed lines and the DSL simulation in dotted lines.







**Figure 11.** Soil temperature at La Cendrosa site (a) and Els Plans (b). Levels comprise at 4, 5, 10, 30 and 35 cm depending on the availability for the site. Observed temperature is shown in solid lines, the NON simulation in dashed lines and the DSL simulation in dotted lines.




**Figure 12.** Parameter sensitivity test for the energy budget terms modifying $K_{dsl}$ and $z_{dsl}$ with a $DSL$ resistance for La Cendrosa site [left] and Els Plans [right] during the day. The color represents the resulting RMSE for the daytime for $Rn$ (a), $LE$ (b), $H$ (c), $G$ (d).





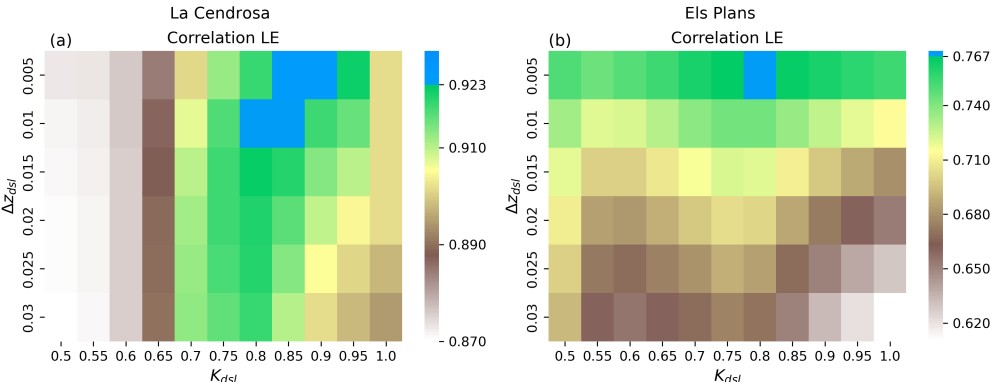

**Figure 13.** Maps of correlation of $LE$ for La Cendrosa (a) and Els Plans (b).

most important since small variations can change the global ET (Swenson and Lawrence 2014) by over 10% as they modulate
the onset of a DSL and the rate of its growth, respectively. Therefore, a sensitivity analysis of $K_{dsl}$ and $z_{dsl}$ was performed for
both sites during the daytime since this is when the ground resistance should have the largest impact on ET. The RMSE of the
estimated fluxes for La Cendrosa (a,c,e,g) and Els Plans (b,d,f,h) for a range of values of $z_{dsl}$ and $K_{dsl}$ during the daytime is
shown in Fig. 12 for the approach with an DSL. Changes in $z_{dsl}$ produce a linear effect as the resistance changes in magnitude,
whereas the effect produced by changes in $K_{dsl}$ is non-linear and it is proportional to the saturation value of the soil.

For $Rn$, the optimal values of RMSE for La Cendrosa and Els Plans in Fig. 12 a and b are in the opposite extremes of the
tested sensitivity values, but the RMSE change is very low, between 4 W m$^2$ for La Cendrosa and 2 W m$^2$ for Els Plans.

For La Cendrosa, the improvement with higher resistance comes from the days after the irrigation period, where the $Rn$ is
overestimated and the decrease in $LE$ is compensated by a rise in $H$ and $G$ together with a reduction of the longwave emission
that reduces the positive bias. For Els Plans, for days when the soil remains dry, the simulation with a larger DSL resistance
value shows a greater $Rn$ compared to simulations with a lower resistance value, whereas under humid conditions, the opposite
is true. The increase in error in $Rn$ with an increasing resistance value depends on several factors and not on a specific period.

For the $LE$ and $H$ fluxes, it can be observed that values of $K_{dsl}$ below 0.6 degrade the simulations compared to the proposed
DSL values of Swenson and Lawrence (2014) (daytime in Tables 5 and 6). In Sect. 5.1, it was observed that although the DSL
simulation produced better results than the simulation without a resistance (NON), the damping of evaporation was greater than
670 optimal for both sites. This is consistent with the sensitivity tests where values of $LE$ with a width resistance of $z_{dsl} = 0.01$ m
show the minimum value of RMSE instead of the value found by Swenson and Lawrence (2014). These discrepancies are
within what would be expected from the LSM dependence on the characterization of other processes and the use of two
contrasting case studies. The error in $H$ is also reduced with a thinner resistance layer. On the other hand, the value of $K_{dsl}$
seems to be optimal at 0.85, closely followed by 0.8, as the error remains within 1 W m$^{-2}$, which corresponds to the proposed
value of Swenson and Lawrence (2014). Zhang et al. (2015) show resistances appearing at values of $K_{dsl}$ close to 0.95 for soils
with coarser texture (e.g. higher sand content), but in soils with finer texture, the resistance is first estimated at lower values



(0.8 to 0.6). The best values for soils with more clay correspond to values of $K_{dsl}$ of 0.9 or higher, although the resistance has a very small value. A value of $K_{dsl}$ close to 0.8 should allow enough open pore space for a DSL to fully form.

The use of a DSL resistance results in the storage of excess energy in the soil, so that the optimum value for $G$ in ISBA corresponds to the lowest possible soil resistance. Other soil constituents and processes can alter the thermal properties of the soil and are still to be included in LSMs (Vereecken et al. 2019). To improve this flux in SURFEX, a more detailed investigation of the thermal properties of the soil is still needed, but this is outside of the scope of the current study.

The correlation values of $LE$ for the values of $K_{dsl}$=0.8 or 0.85 and $z_{dsl} = 0.1$ m are shown in Fig. 13. The correlation for La Cendrosa is improved over the S92 simulation (daytime in Table 5) and for Els Plans it is equal to or slightly better than the NON simulation (daytime in Table 6). This implies optimal values have a greater resistance than S92 but less than that found in Swenson and Lawrence (2014),for the two sites and specific conditions considered in the current study.

## 6  Conclusions

Estimation of evapotranspiration (ET) in semi-arid environments is continuously improving in LSMs, with ET partitioning being a challenge as there are still few measurements characterizing the components of ET. In this context, two contrasting LIAISE sites were investigated. The default simulations showed an overestimation of ET due to the overestimation of bare soil evaporation, a feature common in multiple LSMs which has been addressed by adding surface resistances of different forms. As a consequence, a soil resistance has been implemented in SURFEX V9 using two options: Sellers 92, a resistance formulation widely used by LSMs that was incorporated in V8, and a newly implemented DSL resistance that more accurately represents the actual physical process modulating bare-soil evaporation. There was an improvement of almost 30% in RMSE in ET at each site and no degradation of $H$ using the DSL resistance approach. There was an impact on $Rn$ and $G$ which changed mainly due to the increased soil heating owing to reduced $LE$.

The sites presented several challenges for the model configuration related to vegetation parametrization and management, such as harvesting and irrigation. The first site consisted in an alfalfa field named La Cendrosa, where a detailed characterization of the crop and its biophysical parameters has been made. Such efforts can be later transformed into an informed decision on approximations on a global scale for similar crop types and climates. The development of alfalfa was represented taking into account a daily evolution of $LAI$ and vegetation height based on observations. The study of the photosynthesis parameters used to configure the simulation has shown that the model is sensitive to the cuticular conductance, although it is not the driving mechanism. Instead, the increase in quantum efficiency and the assimilation parameter are most responsible for the increase in ET for alfalfa. It is considered that these measurements and its parameterization in the model improved the estimation of the transpiration. Using a relatively accurate parameter set for photosynthesis and without the presence of the crop after cutting, the DSL resistance becomes important in maintaining the correct amount of bare soil evaporation. The daytime ME and RSME for $LE$ were both reduced by approximately $30\,\mathrm{W\,m^{-2}}$ and the correlation increased from 0.87 to 0.92 when using a DSL, whereas the improvement on $H$ and $Rn$ is of a change in ME and RMSE together of $8\,\mathrm{W\,m^{-2}}$, the same order as the degradation of $G$.



The second site corresponds to a rain-fed natural dry grassland named Els Plans. An almost continuous DSL resistance together with a very low $LAI$ maintains the very low $LE$ fluxes which is consistent with those observed by the eddy covariance system. The differences for $LE$ in the rate of drying after each rain event point to other processes being involved such as wind speed or air saturation. The DSL resistance produced a similar improvement to the La Cendrosa site, but with a reduction in the correlation of $LE$. Two factors played a role: the limitations of a low $LAI$ to characterize vegetation at Els Plans site, and a resistance value that does not take into account the internal biases of the ISBA model for both sites, which led to the need for sensitivity tests.

The parameter sensitivity analysis for the DSL resistance approach suggests a slightly lower $z_{dsl}$ value for the two sites used in the current study than that found in Swenson and Lawrence (2014). The correlation of $LE$ increased by 0.01 for La Cendrosa compared to the initial value of $z_{dsl}$. The correlation with the new values compared to the initial NON simulation was increased by 0.02 at La Cendrosa and is left neutral for Els Plans. The overall error of the simulation was also reduced. The resistance value from the DSL approach was still greater than that obtained when using the S92 method. The point at which resistance begins to develop has been found to be when about twenty percent of the available space in the first layer has been occupied by air.

In addition, the analysis of the two sites has also considered their soil properties, as the DSL involves changes in soil temperature and soil water content. The increase in soil temperature is greater when moisture and bare soil evaporation are significant and the DSL is still present. Laboratory based soil hydraulic properties were available but were found to be insufficient to reproduce the $VWC$, so it is recommended that further studies include variables that define the soil water retention characteristics when calibrating the model.

Finally, the DSL parametrization provides a plausible physical interpretation of the simulated evaporation which is lower than in the baseline scheme while remaining pragmatic since the equation does not represent water vapor transport explicitly. We have shown that it can be applied to conditions that represent the extremes of a semi-arid environment and under different land management practices, including flood irrigation. A multi-site or a global/regional analysis should help to define the choice of parameters for more climates and land cover types. Values with a $K_{dsl}$ between 0.8 and 0.85 and a $z_{dsl}$ near 0.01 m were found for the two sites used in the current study remaining near 0.8 and lower while close to the 0.015 m of Swenson and Lawrence (2014). The DSL resistance methodology seems to be rather robust since similar parameter values are obtained between two different LSMs and varying surface conditions.

*Code and data availability.* SURFEX is an open source code available at https://www.umr-cnrm.fr/surfex/. The modified routines, the SURFEX version code used and simulations can be found at Martí and Boone (2025). The developed DSL option will be included in following versions of SURFEX. The datasets from the LIAISE campaign are available at https://liaise.aeris-data.fr/page-catalogue/ and have been cited in the text. Additional information of the campaign activities can be found in https://www.hymex.fr/liaise/index.html



*Author contributions.* B. Martí carried out the development of SURFEX, the preparation of the forcings, the execution and analysis of the simulations and most of the writing. Jannis Groh participated in the data generation and discussion of the soil properties and reviewed and edited the text. Guylaine participated in the data generation and discussion of the surface energy budget data and reviewed the text. A. Boone supervised and advised the work, reviewed and edited the text, provided the necessary funding for the project, and reviewed and edited the text.

*Competing interests.* The authors declare that they have no conflict of interest.

*Acknowledgements.* We acknowledge Hugo de Boer and Raquel González-Armas for the discussion maintained on A-gs parametrization during the early conception of the article. We acknowledge Oscar Hartogensis and the team of the Wageningen University & Research (WUR) for the alfalfa vegetation measurements. We acknowledge the teams of the *Centre National de Recherches Météorologiques* (CNRM), the UK Metoffice for the measurements of their energy budget stations. We wish to acknowledge the financial support of the French National
Agency for research (grant number ANR-19-CE01-0017 in support of the project "HILIAISE : Human imprint on Land surface Interactions with the Atmosphere over the Iberian Semi-arid Environment")



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
