# Peer review of "Implementation of a dry surface layer soil resistance in two contrasting semi-arid sites with SURFEX-ISBA V9.0"

_EGUsphere, 2025_

## Referee Comment (RC1)

**Implementation of a dry surface layer soil resistance in two contrasting semi-arid sites with SURFEX-ISBA V9.0**

Belén Martí, Jannis Groh, Guylaine Canut, and Aaron Boone.

**Comments to the author.**

This paper explores how adding a dry surface layer (DSL) soil resistance parametrization to the ISBA land surface model (LSM) within the SURFEX V9.0 within the MEB scheme improves the simulation of evapotranspiration (ET) in semi-arid environments. The study focuses on two contrasting field sites from the LIAISE campaign in Spain. The study addresses a longstanding error found in several land surface models, namely an overestimation of evapotranspiration in semi-arid environments, and is an important advance in SURFEX-ISBA parametrization development. A comprehensive evaluation of the DSL scheme is presented with encouraging and interesting results demonstrated. The commentary is well written, succinct (although it is a lengthy manuscript) and meets the objectives laid out in the study.

I recommend this article for publication after addressing a few minor points which are detailed below.

**Minor Revisions**

1. The author has a significant number of parameters which are defined in Section 2. The author could consider a table of parameter lists (including symbol, units, parameter name, and equation number) either in Section 2 or as an Appendix. I feel this would be a useful reference/look-up to help with interpretability of this section.
2. Line 257-259 The SEB stations and associated data are contributions from a number of research groups. The one reference here (Price 2023) is not sufficient. Please add more references for the datasets.
3. Line 286-287 'energy budget of a short dry-down period near the end of the LOP shows a lower Rn compared to La Cendrosa' Would shortwave differences between July & September not also be a factor in the differences in RN between the sites?
4. Line 286 'Short dry-down' Could you add a line about the rainfall event timing that causes this dry-down. When I get to Fig 6, it looks like there is 1 mm rainfall on 02/09, is this sufficient to be termed a dry-down?
5. Section 4.2 What are the lengths of the two simulations (i.e. start/end date) or are they the same as the plots? For clarity, please add a sentence to explain.
6. Figures. There is inconsistency between the date formatting on plots e.g. Fig2/Fig 6 have 07/05 and 05/07. Please could all plots have same start/end wherever possible, this makes it easier to line up harvest dates/irrigation dates and see the impact of these events on various parameters. Please could the lines for harvest/irrigation be added to all relevant plots e.g., Fig 9 and Fig 11.
7. Table 5/6. I think a lot of the statistics displayed in the tables could be included within captions in subplots on Fig4/5. This would make it easier to move between the text and figures and not also have to refer to the tables. The tables could again be moved into an Appendix.
8. Table5/6 It is not clear which parameter the 'Corr' relates to - it could be H/LE/G from the way the table is laid out. I have assumed it is H and LE, but it is not clear.
9. Line 451 'low/high vegetation' I think this is the first time you use this – could you add a line to explain what is meant by this.
10. Line 463 'Cendrosa (56Wm−2) and Els Plans (50Wm−2)' I struggled to see where these numbers came from in Table 5/6. Could you please check.

11. Line 474 'except for an improvement from 0.81 to 0.88 in the overall correlation' I could be wrong, but are these not the correlations for La Cendrosa, whilst the Els Plans correlations are 0.87-0.89?
12. Line 476 The value of 57 Wm-2 isn't in Table 6 for the G parameter. Please check and consistency within text.
13. Line 486 Please add the date of the two irrigation events in brackets.
14. Line 485-490 Please could you add a short explanation as to why the LE differences are so different after the two irrigation events. This is presumably due to the high/low vegetation differences, but it would be good to make this clear to readers.
15. Fig 6 The very low precipitation is quite difficult to see, particular in (b). Can the second plot axis be changed to make the precip clearer.
16. Fig 6 Please add the precipitation into the legend (red) obs precip (black) obs LE.
17. Fig 7 Please consider having the same scale for Rsoil for (a) and (b) as this will highlight the different magnitudes of the resistance between the two sights.
18. Fig 9 Either in the manuscript text or in the Figure caption, please could you provide a line or two what the standard deviation is from the simulation? Is this the standard deviation of the three simulations? Or is this one simulation (if so, which)?
19. Fig 9 Please could you include additional date ticks on panel (a).

**Technical and typographical corrections**

Line 54-56 Unclear sentence, please consider revising.

Line 83 Please define SEB

Line 121 'where f indicates the proportionality of internal CO2 and inside the leaf boundary layer' remove and?

Line 216 Reference formatting incorrect.

Line 256 Change 'part' to 'region'

Fig 2/ Fig 3 – Spacing needed between theRes.

Line 369 Typo

Line 443 Typo? – residual

Line 478 Typo? Solids > Soils?

Line 477–480 Check sentence structure

Line 489 Please change 'other irrigation event' to 'second irrigation event', and please include the date of the event in brackets

Line 541-542 Date formatting: 29 July

---

## Referee Comment (RC2)

Review of the manuscript entitled "Implementation of a dry surface layer soil resistance in two contrasting semiarid sites with SURFEX-ISBA V9.0" by Belen Marti et al. This manuscript develops a new dry surface layer (DSL) resistance for the ISBA LSM to simulate more realistically evapotranspiration processes in rural areas. Two observational sites located in the north-eastern of Spain are used to validate the performance of the model. Three simulations (offline) are performed with the LSM in each site, one run does not use a soil resistance parameterization (hereafter NON simulation), one experiment uses the DSL approach (henceforward DSL simulation), and finally, the third simulation considers the soil resistance parameterization developed by Sellers et al. (1992b and 1996), hereafter S92 simulation. Modeled sensible, latent, and net radiative heat fluxes are compared against observations and common statistical errors are calculated for each experiment and site. This manuscript is interesting but considerable changes are needed before it can be accepted for publication.

1. Abstract, line 4. Before "models" add "land surface".

2. Page 3, line 75. "Both parametrizations are tested", what parametrizations? The only parametrization described until now is the dry surface layer (DSL).

3. Section 2 should be reduced considerably. If you think all equations are important add them in an appendix or as a supplementary document.

4. Section 3.1. All the observational data collected in both sites and used for validating the simulations should be clearly explained.

5. Section 3.2. How were the SEB terms calculated/observed, especially the ground heat flux (G)? In general, the ground heat flux is calculated as the residual term of the SEB equation, that is, $G = Rn - H - LE$. However, in this study/experiment is not the case. Please, clarify.

6. Section 3.3, line 288. The fact that the albedo is lower in El Plans than in La Cendrosa cannot explain the lower value for the Rn. Low albedos increase net radiative heat fluxes.

7. Section 4.1, line 305. "The wind speed was filled with the 3 m data at the same site", It is the first time that observations are reported at this height in the manuscript. Was the wind speed recorded at 10 m or 3 m above ground? or at both heights?

8. Section 4.2.1. Is this large section needed? A summary with a table showing the values used in the runs should be sufficient. Results section starts at page 17!!!

9.  Section 5.1. Why do you think the ground heat flux is not improved with the new dry surface layer approach? How is this term calculated in the LSM?

10. Section 5.1, line 476. I was not able to find in Table 6 the daily RMSE reported for the ground heat flux in the manuscript (i.e., 57 W/m$^2$).

11. Figs 4-5. The period analyzed in these figures should be included in the captions.

12. Figs. 6, 7, and 8. The format of the days simulated (DD/MM) is not adequate.

13. Fig. 7b. Why does Fig. 7b show results for three months and Fig. 6b only shows results for 15 days for the Els Plans site?

14. Section 5.2.1. The resistance values shown in Fig. 7 are very different between S92 and DSL simulations, why do you think these significant differences are not producing substantial latent heat fluxes disparities?

15. Section 5.3.1. Is this section needed?

16. Section 5.3.2, Fig. 10a. It seems to me (based on Fig. 10a) that the VWC is better captured by the NON simulation than by the DSL run, especially after the ~ 11$^{th}$ and before the ~ 24$^{th}$ of July, could you explain why?

17. Section 5.3.2, Fig. 10. The format of the days simulated (DD/MM) is not adequate.

18. Section 5.3.2, Fig. 10b. Again, the VWC seems better simulated by the NON experiment than by the DSL simulation, could you explain why?

19. Section 5.3.3. Fig. 11a shows that the NON simulation reproduces considerably better the maximum soil temperature at 5 cm (below the ground surface) than the DSL simulation for an approximately a 10-day period in the middle of the month (July), could you explain why? Also, the DD/MM format is not adequate here.

20. Section 5.4, line 661. How were the RMSE values of 4 and 2 W m$^{-2}$ reported here calculated?

---

## Author Comment (AC1)

**Review 1**

Implementation of a dry surface layer soil resistance in two contrasting semi-arid sites with

SURFEX-ISBA V9.0

Belén Martí, Jannis Groh, Guylaine Canut, and Aaron Boone.

Comments to the author.

This paper explores how adding a dry surface layer (DSL) soil resistance parametrization to the ISBA land surface model (LSM) within the SURFEX V9.0 within the MEB scheme improves the simulation of evapotranspiration (ET) in semi-arid environments. The study focuses on two contrasting field sites from the LIAISE campaign in Spain. The study addresses a longstanding error found in several land surface models, namely an overestimation of evapotranspiration in semi-arid environments, and is an important advance in SURFEX-ISBA parametrization development. A comprehensive evaluation of the DSL scheme is presented with encouraging and interesting results demonstrated. The commentary is well written, succinct (although it is a lengthy manuscript) and meets the objectives laid out in the study.

I recommend this article for publication after addressing a few minor points which are detailed below.

We thank you for your positive review. We write this response in the past tense for clarity reasons. The modifications you suggest have been implemented as described in this response, the suggestions by the other reviewers have also been taken into account. The main text has been considerably reduced by moving the vegetation model description to an appendix, but other reviewer comments have asked for some additional text, although mostly minor.  The main changes that the revised manuscript has following reviewer's comments are:
-Part of section 2.3.1 has been moved to the introduction by another review's suggestion
-Reduction on the number of equations, moving the larger part of the vegetation model description to an Appendix
-Tables that can be omitted from the main text have been moved to the appendices.
-Addition of a glossary (symbols, units: as suggested by the review)
-The vegetation parameter discussion has been moved to an appendix and only the description of parameter value choices is left on the main text due to the extension of the main text.
-Small modifications to the figures as per the review
-A section to better prepare the reader for the sensitivity analysis has been added
-A table of sensors of the surface energy stations has been added and together with further citations of available related measurements
-The description of the measurements available has been clarified

Minor Revisions

1. The author has a significant number of parameters which are defined in Section 2. The author could consider a table of parameter lists (including symbol, units, parameter name, and equation number) either in Section 2 or as an Appendix. I feel this would be a useful reference/look-up to help with interpretability of this section.

An Appendix including the list of parameters, in the form of a glossary, has been added. It has been cited in 2.1.

2. Line 257-259 The SEB stations and associated data are contributions from a number of research groups. The one reference here (Price 2023) is not sufficient. Please add more references for the datasets.

The other citations available in the Aeris platform that houses the LIAISE database have been added in the following way:

"The field experiment took place in the north-eastern region of the Iberian Peninsula from April 2021 to the end of September 2021 (the Long Observational Period, LOP). Surface energy budget stations were installed over alfalfa (Canut 2022a; Mangan 2022), maize (Martínez-Villagrasa 2022), irrigated grass (Miró 2021), vineyard, apple orchard, almond orchard (Canut 2022b) and natural rainfed grass (Price 2023). "

3. Line 286-287 'energy budget of a short dry-down period near the end of the LOP shows a lower Rn compared to La Cendrosa' Would shortwave differences between July & September not also be a factor in the differences in RN between the sites?

Yes, it is a relevant factor. The downwelling shortwave radiation at Els Plans decreases from a diurnal peak of almost 1000 W m-2 to 800 W m-2 between July and September. La Cendrosa downwelling shortwave radiation is around 950  W m-2 during the period of growth shown.  To reflect this, the text in italics below has been modified, this paragraph is also affected by comment 4 below. The text reads as:

"The energy budget of a short dry-down period near the end of the LOP (Fig. 3) shows a lower Rn compared to La Cendrosa *with two small rain contributions of 2 mm and 0.8 mm on the eve of 2 September. The Rn difference is due to the contribution of the net long wavelength radiation being lower and the time difference between the two periods which can account for up to 100* W m-2."

4. Line 286 'Short dry-down' Could you add a line about the rainfall event timing that causes this dry-down.

We have added to the revised manuscript the text in italics to the following paragraph:

"The Els Plans site is a rainfed, relatively dry area with natural grass that was drying during the LOP. The parcel is located within a special protection area for steppe birds, and it is not cultivated. The energy budget of a short dry-down period near the end of the LOP is shown (Fig. 3) *with two small rain contributions of 2 mm and 0.8 mm on the eve of 2 September. The Rn difference is due to the contribution of the net long wavelength radiation being lower and the time difference between the two periods which can account up to 100* W m-2."

When I get to Fig 6, it looks like there is 1 mm rainfall on 02/09, is this sufficient to be termed a dry-down?

According to McColl et al. (2017) and Akbar et al. (2018), a dry-down is defined as follows: it starts after the last rainfall event, with no rainfall on the first day, and rainfall on subsequent days should not exceed 1 mm per day. During the dry-down, soil moisture should decline, and it should last at least 5 days. The previous choice of time axis did not show the full precipitation. We have now modified it so it is visible. The precipitation range has also been modified in this figure at the request of another reviewer. Although a precipitation event occurred the night of September 9 it is smaller than 1mm, and it wasn't the intended target of the discussion. The increase of VWC for the following 5 days is also less than 1%. Thus, the period after the rainfall event on 02/09 the period shown in the figure can be defined as dry-down.

[Figure]

5. Section 4.2 What are the lengths of the two simulations (i.e. start/end date) or are they the same as the plots? For clarity, please add a sentence to explain.

The simulation of la cendrosa starts 1 July at 00:00 UTC and finishes 1 August at 00:00 UTC. The simulation of Els plans 17 June 10:00 UTC to 29 September at 09:00 UTC. Some plots have a reduced time length due to the variable characteristics. The text in italics has been added to the reviewed document:

"The simulations of La Cendrosa and Els Plans are performed offline (i.e. driven by observations as input). *They comprise the periods from 1 July at 00:00 UTC to 1 August at 00:00 UTC and from 17 June 10:00 UTC to 29 September at 09:00 UTC respectively.* The associated atmospheric variables are the incident short and longwave radiation fluxes, wind speed, temperature, specific humidity, pressure, atmospheric $CO_2$ concentration, and rainfall rate at a 30 minute time step."

6. Figures. There is inconsistency between the date formatting on plots e.g. Fig2/Fig 6 have 07/05 and 05/07.

We have corrected formatting so it is consistent between the plots you mention and the following plots. We keep the format MM/DD. Thank you for the detailed revision.

Please could all plots have same start/end wherever possible, this makes it easier to line up harvest dates/irrigation dates and see the impact of these events on various parameters.

The simulation of La Cendrosa covers only the month of July and the simulation of els Plans is much longer at almost 4 months. Due to the electrical problems affecting the measurements at Els Plans, however, very few dry-down periods were captured. The current choice of limits needs to be maintained as is for the indicated previous reasons.

Please could the lines for harvest/irrigation be added to all relevant plots e.g., Fig 9 and Fig 11.

They have been added to Figures 9 and 11. In Figure 10, the irrigation lines have been added in the background so they don't cover the VWC, as the increase of this variable coincides with the irrigation event. The rest of the figures have the pertinent lines.

7. Table 5/6. I think a lot of the statistics displayed in the tables could be included within captions in subplots on Fig4/5. This would make it easier to move between the text and figures and not also have to refer to the tables. The tables could again be moved into an Appendix.

The ME, RMSE for all fluxes for the daily values are included within the figure frame as shown for Fig. 4. Since we cite within the text the daily values, the daytime values and the night values it is preferred to keep the tables on the main text.

[Figure]

The following text is added to the captions:

*Daily mean error and root mean square error are included in the figure frame.*

8. Table 5/6 It is not clear which parameter the 'Corr' relates to - it could be H/LE/G from the way the table is laid out. I have assumed it is H and LE, but it is not clear.

The caption has been clarified in the revised manuscript, for example for Els Plans:

"Mean Error (ME) and Root Mean Square Error (RMSE) in W m-2 for the net radiation (Rn), the sensible heat flux (H), the latent heat flux (LE), and the ground heat flux (G) for Els Plans site. *The correlation (Corr) between the simulations and the observations has been included for H and LE.* The values taking into account the residual using the Bowen ratio method are indicated in brackets."

The header of the tables indicates the variable the correlation to which it corresponds.

9. Line 451 'low/high vegetation' I think this is the first time you use this – could you add a line to explain what is meant by this.

It has been clarified as:

*In order to describe the parameter selection methodology, we provide a description of the low vegetation scheme in ISBA (Calvet 2000), which is used to represent crops and herbaceous types as opposed to high vegetation that considers woody types. The main difference is in the relationship between the mesophilic conductance and the maximum humidity deficit.*

10. Line 463 'Cendrosa (56Wm−2) and Els Plans (50Wm−2)' I struggled to see where these numbers came from in Table 5/6. Could you please check.

These values are an additional calculation, not a comparison between simulation and observation, as they correspond to the residue and not the fluxes. We have added the values of the fluxes statistics to reduce confusion and changed the verb from 'observed' to 'calculated here' in order to clarify they are not in the table.

"The improvement of the statistics is also present for H, *with* the underestimation *being* reduced*, and the RMSE reducing to 37 W m-2 for La Cendrosa and 45 W m-2 for Els Plans These values are* within the values of the standard deviation observed *calculated here* for the residue for La Cendrosa (56 W m-*2*) and Els Plans (50 W m-*2*)."

11. Line 474 'except for an improvement from 0.81 to 0.88 in the overall correlation' I could be wrong, but are these not the correlations for La Cendrosa, whilst the Els Plans correlations are 0.87-0.89?

You are right, the values are not those of Els Plans, they have been corrected in the revised manuscript.

12. Line 476 The value of 57 Wm-2 isn't in Table 6 for the G parameter. Please check and consistency within text.

It should be 70 W m-2: the value was read erroneously from the table of la Cendrosa instead of Els Plans, indeed it is a typo. It has been corrected in the revised manuscript. We thank you for the fact check. We have revised the text for similar errors.

13. Line 486 Please add the date of the two irrigation events in brackets.

This has been done in the revised manuscript, see next comment.

14. Line 485-490 Please could you add a short explanation as to why the LE differences are so different after the two irrigation events. This is presumably due to the high/low vegetation differences, but it would be good to make this clear to readers.

We have added the sentence in italics:

The observed LE is shown for the aforementioned simulations in Fig. 6a together with a top to bottom barplot with rain and irrigation. The largest change between simulations is after the first irrigation event *[July 11]*. The increase of observed LE is overestimated with the NON option, mainly due to the large contribution from soil evaporation, as in Lohou et al. (2014), whereas the LE in the DSL simulation is reduced more rapidly, which matches more accurately the observations in this period. The *second* irrigation event *[July 24]* and small rain events result in small increases

in LE that decrease during the ensuing hours since the soil is still near saturation. *The difference in the maximum LE between irrigation events is due to the change in transpiration. The first event occurs during very low vegetation whereas the second corresponds to fully grown alfalfa.*

15. Fig 6 The very low precipitation is quite difficult to see, particular in (b). Can the second plot axis be changed to make the precip clearer.

We have changed the axis for figure b, from 0 to 10. See the new graphic below.

[Figure]

16. Fig 6 Please add the precipitation into the legend (red) obs precip (black) obs LE.

This change has been made in the revised manuscript. See comment 15.

17. Fig 7 Please consider having the same scale for Rsoil for (a) and (b) as this will highlight the different magnitudes of the resistance between the two sights.

Figure 7b has been modified to have the same Rsoil axis as Figure 7a.

[Figure]

18. Fig 9 Either in the manuscript text or in the Figure caption, please could you provide a line or two what the standard deviation is from the simulation? Is this the standard deviation of the three simulations? Or is this one simulation (if so, which)?

It corresponds to the standard deviation of the 30 minute time series of the observed and simulated albedo. It comes from the NON simulation but all three coincide as albedo is imposed and the LAI and vegetation height which could modify it also coincide.

The following sentence in italics has been added:

The observed daily variability of albedo and its model counterpart are shown for La Cendrosa in Fig. 9a. *The standard deviation shown is from the 30 minute time series of the observations and the simulation.*

Additionally, the caption in the revised manuscripts now:

"Figure 9: Observed (blue) and simulated (red) albedo for La Cendrosa (a) and Els Plans sites (b) for the NON simulation. Each boxplot measurement shows a central line with the median value. The size of the boxes correspond to the quartiles of albedo observations within the day and the error bars represent the variability of the albedo within the day."

19. Fig 9 Please could you include additional date ticks on panel (a).

This has been done in the revised manuscript. Vertical lines have also been added.

[Figure]

Technical and typographical corrections

Line 54-56 Unclear sentence, please consider revising.

The sentence has been split and modified as:

" Transpiration is modeled using a resistance analogue together with the humidity gradient. The resistance represents the canopy biophysical and turbulent processes. One of the configurations includes the assimilation of carbon, which allows the specification of species-dependent plant parameters."

Line 83 Please define SEB
It has been replaced in the revised manuscript.

Line 121 'where f indicates the proportionality of internal CO2 and inside the leaf boundary layer' remove and?
It has been replaced in the revised manuscript

Line 216 Reference formatting incorrect.

The format has been revised for this reference and all others

Line 256 Change 'part' to 'region'
It has been replaced in the revised manuscript

Fig 2/ Fig 3 – Spacing needed between theRes.

It has been replaced in the revised manuscript

Line 369 Typo
It has been replaced in the revised manuscript

Line 443 Typo? – residual

Intentional to have residue

Line 478 Typo? Solids > Soils?

It is solids, such as quartz. The sentence is clarified adding 'in the soil', see comment below.

Line 477–480 Check sentence structure

The sentence has been split into two. It has been written as:

"A sensitivity test indicated that reducing the thermal conductivity of solids in the soil by 25% could reduce this error by reducing the G flux up to 20W m^-2, while increasing the error in the H proportionally. This value is within the range of different soil types reviewed from observations by Zhang et al. (2017)"

Line 489 Please change 'other irrigation event' to 'second irrigation event', and please include the date of the event in brackets
It has been replaced in the revised manuscript

Line 541-542 Date formatting: 29 July

Modified, two other dates have been changed too
—--------------------------------------------------

Bibliography

Akbar, R., Short Gianotti, D. J., McColl, K. A., Haghighi, E., Salvucci, G. D., & Entekhabi, D. (2018). Estimation of landscape soil water losses from satellite observations of soil moisture. *Journal of Hydrometeorology*, *19*(5), 871-889.

Aouade, G., Jarlan, L., Ezzahar, J., Er-Raki, S., Napoly, A., Benkaddour, A., Khabba, S., Boulet, G., Garrigues, S., Chehbouni, A., et al.: Evapotranspiration partition using the multiple energy balance version of the ISBA-Ag s land surface model over two irrigated crops in a semi-arid Mediterranean region (Marrakech, Morocco), Hydrology and Earth System Sciences, 24, 3789–3814, 2020.

Calvet, J.-C.: Investigating soil and atmospheric plant water stress using physiological and micrometeorological data, Agricultural and Forest Meteorology, 103, 229–247, 2000.

Canut, G.: LIAISE_LA-CENDROSA_CNRM_MTO-FLUX-30MINL2[Dataset]Aeris, https://doi.org/10.25326/320, 2022a.

Canut, G.: LIAISE_PREIXANA_CNRM_MTO-FLUX-30MINL2[Dataset]Aeris, https://doi.org/10.25326/361, 2022b.

Mangan, M., Hartogensis, O., Branch, O., Martinez Villagrasa, D., Boone, A., Canut, G., Cuxart, J., de Boar, H., Le Page, M., Miro, J., Price, J., and Vila Guerau de Arellano, J.: LIAISE_UNIFIEDEC_WUR_10MIN_L1 [Dataset] Aeris, https://doi.org/10.25326/389, 2022

Martínez-Villagrasa, D., M., B., Cuxart, J., and Wrenger, B.: LIAISE_IRTA-CORN_UIB_SEB-10MIN_L2 [Dataset] Aeris, https://doi.org/10.25326/344, 2022

McColl, K. A., Wang, W., Peng, B., Akbar, R., Short Gianotti, D. J., Lu, H., ... & Entekhabi, D. (2017). Global characterization of surface soil moisture drydowns. *Geophysical Research Letters*, *44*(8), 3682-3690.

Miró, J. R.: LIAISE_IRTA-ET0_SMC_SEB-10MN_L1 [Dataset] Aeris, https://liaise.aeris-data.fr/page-catalogue/?uuid=10007f83e709-4ed4-49f1-b26f-c45d0519e4cf, 2021

Sobaga, A., Decharme, B., Habets, F., Delire, C., Enjelvin, N., Redon, P.-O., Faure-Catteloin, P., and Le Moigne, P.: Assessment of the interactions between soil–biosphere–atmosphere (ISBA) land surface model soil hydrology, using four closed-form soil water relationships and several lysimeters, Hydrology and Earth System Sciences, 27, 2437–2461, 2023.

Price, J.: LIAISE_ELS-PLANS_UKMO_MTO-30MINL2[Dataset]Aeris, https://doi.org/10.25326/430, 2023

Weber, T. K. D., Weihermüller, L., Nemes, A., Bechtold, M., Degré, A., Diamantopoulos, E., Fatichi, S., Filipović, V., Gupta, S., Hohenbrink, T. L., Hirmas, D. R., Jackisch, C., de Jong van Lier, Q., Koestel, J., Lehmann, P., Marthews, T. R., Minasny, B., Pagel, H., van der Ploeg, M., Shojaeezadeh, S. A., Svane, S. F., Szabó, B., Vereecken, H., Verhoef, A., Young, M., Zeng, Y., Zhang, Y., and Bonetti, S.: Hydro-pedotransfer functions: a roadmap for future development, Hydrol. Earth Syst. Sci., 28, 3391–3433, https://doi.org/10.5194/hess-28-3391-2024, 2024.

Jackisch, C., Germer, K., Graeff, T., Andrä, I., Schulz, K., Schiedung, M., Haller-Jans, J., Schneider, J., Jaquemotte, J., Helmer, P., Lotz, L., Bauer, A., Hahn, I., Šanda, M., Kumpan, M., Dorner, J., de Rooij, G., Wessel-Bothe, S., Kottmann, L., Schittenhelm, S., and Durner, W.: Soil moisture and matric potential – an open field comparison of sensor systems, Earth Syst. Sci. Data, 12, 683–697, https://doi.org/10.5194/essd-12-683-2020, 2020.

Zhang, N. and Wang, Z.: Review of soil thermal conductivity and predictive models, International Journal of Thermal Sciences, 117, 172–183, 2017.

---

## Author Comment (AC2)

General comments

Accurate estimation and partitioning of evapotranspiration (ET) are critical for understanding land–atmosphere interactions, yet current land surface models (LSMs) still exhibit notable deficiencies in representing transpiration and evaporation, particularly soil evaporation. In this study, the authors introduce a dry surface layer (DSL) resistance parameterization into the ISBA model and conduct simulations and evaluations at two contrasting semi-arid sites, leveraging comprehensive observational data from the LIAISE campaign. Their results demonstrate that the new DSL parameterization effectively reduces bare soil evaporation and helps address existing shortcomings in ET modeling, offering valuable insights for improving LSM performance in estimating evapotranspiration.

The manuscript is well-organized, with a robust methodological approach and substantial supporting data. The logical structure and discussion are also appropriate. However, the text is somewhat lengthy, and there is room for improvement in both language clarity and figure presentation. Revisions are necessary before the manuscript can be considered for publication.

We thank you for your positive review. We write this response in the past tense for clarity reasons. The modifications you suggest have been implemented as described in this response, the suggestions by the other reviewers have also been taken into account. The main text has been considerably reduced by moving the vegetation model description to an appendix, but other reviewer comments have asked for some additional text, although mostly minor. The main changes that the revised manuscript has following reviewer's comments are:
-Part of section 2.3.1 has been moved to the introduction by another review's suggestion
-Reduction on the number of equations, moving the larger part of the vegetation model description to an Appendix.
-Tables that can be omitted have been moved to the appendices.
-Addition of a glossary (symbols, units: as suggested by the review process)
-Vegetation parameter discussion is moved to an appendix and only parameter value decision is left on the main text.
-Small modifications to the figures as per the review process
-A section to prepare the reader for the sensitivity analysis has been added
-A table of sensors of the surface energy stations has been added and together with further citations of available related measurements
-The description of the measurements available has been clarified

The estimation of soil evaporation and DSL resistance is influenced by soil state variables such as soil moisture and soil temperature. However, the simulation results of these variables presented in

this study (Figures 10 and 11) appear to be unsatisfactory, which raises concerns about the reliability of subsequent model evaluation outcomes.

For the VWC, the problems are present in both NON and DSL simulations. The problem was already acknowledged in the conclusions. This problem is not DSL dependent but rather pedo-transfer function dependent. A citation to emphasize it has been added since such results have been reported before (Aouade et al. , 2020) 'Laboratory based soil hydraulic properties were available but were found to be insufficient to reproduce the VWC, *a behaviour also reported in Aouade et al. (2020).* It is recommended that further studies include variables that define the soil water retention characteristics when calibrating the model.'

Regarding reliability, the issues of the VWC were reported and Sobaga et al. (2023) study was pointed as a future improvement compared to our current results. Additionally, as other reviewers still had some doubts on the VWC section, clarification between the NON and DSL simulations for the VWC has been added.

The temperature differences have been reported explicitly in the submitted article and the text points to a partial solution modifying the soil heat capacity. This is done in the original article of Swenson and Lawrence (2014) with the change of thickness of the first soil layer. Since the measurements of the soil properties did not include the thermal capacity we preferred to leave the default values so the reader can make the right assessment of the changes that the soil resistance induces. The increment in temperature of the S92 resistance while smaller by about half of that of the DSL, it is also present. The article showing the increase of temperature provides the full picture of the impact of resistance values which are higher than those published before. We had already declared in the conclusions that the resistance needs further testing in the model at a global scale, and this article gives the necessary perspective of key variables for a follow up study.

We made an effort to report on all variables in order to help identify the issues that can appear. In addition, we also provided suggestions in the text of how to approach them, such as testing the new pedo-tranfer equations or modifying the soil heat capacity for the soil temperature. The original article of Swenson and Lawrence (2014) does not include these variables, acknowledging these limitations for which improvements in these functions is a key for future use of this resistance.

It is recommended to reorganize the introduction and Sect. 2.3.1 for improved clarity. Currently, the Introduction does not provide sufficient explanation of specialized terms such as soil resistance, particularly dry surface layer (DSL) resistance. As these concepts may not be familiar to the broader readership of GMD, it would be beneficial to consider moving part of the description of soil resistance from Sect. 2.3.1 into the Introduction.

Section 2.3.1 has been reduced in the revised manuscript and part of the information moved to the introduction. The review of the literature has been moved to the introduction together with the importance of VWC for soil resistances and the data used to calibrate the DSL.

Sect. 5.4 presents the results of the sensitivity analysis. However, it would be helpful to briefly introduce the concept of sensitivity analysis and the specific method adopted in this study in the earlier methodological sections. This would provide clearer context and improve the continuity of the manuscript.

A new subsection has been added at the end of section 2 with the following text:

The DSL configuration for the main comparison is the one of Swenson and Lawrence (2014) but its parameters may not be the most suitable for ISBA. A sensitivity analysis, consisting in the variation of two parameters for the DSL resistance is later presented. This kind of analysis is necessary to characterize the changes of output variables and diagnose them. The response to the change can be identified, whether it is linear, nonlinear or negligible. As the number of parameters increases, the behaviours of outputs become intertwined and cannot be necessarily easily predicted. The sensitivity analysis identifies whether a parameter is relevant for a certain variable. We generate multiple simulations with two varying parameters and use the root mean square error (RMSE) of the simulations to suggest the more appropriate values for the estimation of the turbulent fluxes with ISBA for the DSL option.

Specific comments

Line 1: Latent heat flux (LE) and evapotranspiration are closely related but conceptually distinct terms, with different physical meanings and units. The manuscript should carefully review its use of both terms to avoid confusion. In particular, Figure 8 incorrectly labels the unit of LE as "mm," which is not appropriate.

We have changed Figure 8 by E in the labels and caption and specify it is accumulated evaporated water in the text, modifying ET when not appropriate.

Line 61-64: It is recommended to number the four drying stages to make it easier for the reader to understand.

They have been numbered now in the revised manuscript

Line 83: The term "surface energy budget (SEB)" appears for the first time here, rather than at line 257.

The term appears now at its first appearance in the revised manuscript

Line 131: I don't quite understand what N and X represent in this context.

N and X represent the two possible strategies, tolerant and resistant. The text has been modified as follows to clarify:

"where D+max can be N for the drought-avoiding strategy applied for C3 crops (eq. A7), and X for the drought-tolerant strategy (eq. A9). *Given the same Dmax, evapotranspiration will decrease for C3 and increase for C4 plants under water stress conditions due to these different strategies.*"

Line 216: Please revise the citation "(Iden et al. 2021)" to "Iden et al., (2021)" to match the correct citation style. Similar formatting inconsistencies may exist elsewhere and should be reviewed.

It is revised for this citation and the other citations have been checked and revised for consistency.

Line 308: For La Cendrosa, irrigation was treated as rainfall by adding 30 mm of water between 00:00 and 02:00 UTC.
The text has been replaced with your proposed sentence in the revised manuscript

Line 323-324: While the alfalfa field is clearly identified as a C3 crop, the grass type used for Els Plans is not specified as either C3 or C4. Given that this distinction may affect model performance, the authors should clarify which type was used. Additionally, in Table 1, the crop is classified under the "C4 crop" category. Please check if this classification is correct.

Crops and grass can be either C4 or C3. The table now indicates crop/herbaceous and the text has been clarified indicating the grass at Els plans is simulated as C3.

Line 330-331: It would be helpful if the manuscript could briefly clarify what the "AST option" within SURFEX refers to.
It has been clarified in contrast to other options. The text now reads as:
"Additionally, the AST option within SURFEX is used for both simulations. With this option, the A-gs scheme is used to model photosynthesis *parameterizing its processes in contrast with other options that model transpiration directly without considering the biological processes.*"

Line 347: "is" to "in".

It is changed now in the revised manuscript

Line 353: "one" to "1".

It is changed now in the revised manuscript

Table 1: It is recommended to unify the formatting of the table and add border lines to both the top and bottom.

It is changed now in the revised manuscript

Figure 9: It is necessary to provide a clear explanation of the meaning of the error bars and the orange horizontal lines shown in the figure.

A description has been added to the caption. It now reads as:
"Figure 9: Observed (blue) and simulated (red) albedo for the NON simulation for La Cendrosa (a) and Els Plans sites (b). Each boxplot measurement shows a central line with the median value. The size of the boxes correspond to the quartiles of albedo observations within the day and the error bars to the variability of the albedo within the day."

Bibliography

Aouade, G., Jarlan, L., Ezzahar, J., Er-Raki, S., Napoly, A., Benkaddour, A., Khabba, S., Boulet, G., Garrigues, S., Chehbouni, A., et al.: Evapotranspiration partition using the multiple energy balance version of the ISBA-Ag s land surface model over two irrigated crops in a semi-arid Mediterranean region (Marrakech, Morocco), Hydrology and Earth System Sciences, 24, 3789–3814, 2020.

Sobaga, A., Decharme, B., Habets, F., Delire, C., Enjelvin, N., Redon, P.-O., Faure-Catteloin, P., and Le Moigne, P.: Assessment of the interactions between soil–biosphere–atmosphere (ISBA) land surface model soil hydrology, using four closed-form soil water relationships and several lysimeters, Hydrology and Earth System Sciences, 27, 2437–2461, 2023.

Swenson, S. and Lawrence, D.: Assessing a dry surface layer-based soil resistance parameterization for the Community Land Model using GRACE and FLUXNET-MTE data, Journal of Geophysical Research: Atmospheres, 119, 10–299, 2014.

---

## Author Comment (AC3)

Review 2

Review of the manuscript entitled "Implementation of a dry surface layer soil resistance in two contrasting semiarid sites with SURFEX-ISBA V9.0" by Belen Marti et al.
This manuscript develops a new dry surface layer (DSL) resistance for the ISBA LSM to simulate more realistically evapotranspiration processes in rural areas. Two observational sites located in the north-eastern of Spain are used to validate the performance of the model. Three simulations (offline) are performed with the LSM in each site, one run does not use a soil resistance parameterization (hereafter NON simulation), one experiment uses the DSL approach (henceforward DSL simulation), and finally, the third simulation considers the soil resistance parameterization developed by Sellers et al. (1992b and 1996), hereafter S92 simulation. Modeled sensible, latent, and net radiative heat fluxes are compared against observations and common statistical errors are calculated for each experiment and site. This manuscript is interesting but considerable changes are needed before it can be accepted for publication.

We thank you for your positive review. We write this response in the past tense for clarity reasons. The modifications you suggest have been implemented as described in this response, the suggestions by the other reviewers have also been taken into account. The main text has been considerably reduced by moving the vegetation model description to an appendix, but other reviewer comments have asked for some additional text, although mostly minor.  The main changes that the revised manuscript has following reviewer's comments are:
-Part of section 2.3.1 has been moved to the introduction by another review's suggestion
-Reduction on the number of equations, moving the larger part of the vegetation model  description to an Appendix
-Tables that can be omitted from the main text have been moved to the appendices.
-Addition of a glossary (symbols, units: as suggested by the review process)
-Vegetation parameter discussion is moved to an appendix and only parameter value decision is left on the main text due to the extension of the main text.
-Small modifications to the figures as per the review process
-A section to prepare the reader for the sensitivity analysis has been added
-A table of sensors of the surface energy stations has been added and together with further citations of available related measurements
-The description of the measurements available has been clarified

1. Abstract, line 4. Before "models" add "land surface".

It has been replaced in the revised manuscript

2. Page 3, line 75. "Both parametrizations are tested", what parametrizations? The only parametrization described until now is the dry surface layer (DSL).

The citation of each parameterization has been added in the previous paragraphs and the line reads as: "Both the Sellers et al. (1992) and the Swenson and Lawrence (2014) parametrizations"

3. Section 2 should be reduced considerably. If you think all equations are important add them in an appendix or as a supplementary document.

This section is interlinked with the vegetation parameterization of the alfalfa, which has been moved to an appendix and reduced to a specification of the parameter values in the main text. Consequently, section 2.1 has conserved the latent heat decomposition and the vegetation decomposition has been moved to the appendix together with all of section 2.2.

4. Section 3.1. All the observational data collected in both sites and used for validating the simulations should be clearly explained.

A paragraph has been added indicating existing measurements and where to find further description at the end of section 3.1:

*Two surface stations were installed at la Cendrosa, the longer series was taken Canut 2022a). Measurements of temperature, humidity and wind at different levels depending on sensor availability and high frequency measurements were available 3 m, 10 m, 25 m and 50 m for the two sites (see Boone et al. (2025) and Brooke et al. (2023) for more details).*

The type of data used was already explained in sections 3.2 and 3.3.  Clarification on the surface energy budget terms has been added as indicated in the following comment.

5. Section 3.2. How were the SEB terms calculated/observed, especially the ground heat flux (G)? In general, the ground heat flux is calculated as the residual term of the SEB equation, that is, G = Rn – H – LE. However, in this study/experiment is not the case. Please, clarify.

A four component radiometer and an eddy covariance measurements with a gas analyser were used for Rn, H and LE respectively. A table of the sensors has been added to the appendix and is cited in the text. A buried flux plate measures the soil temperature and the soil temperature measurements allow the use of the calorimetric method to correct for the storage and adapt the flux to represent the flux of energy into the soil at the surface.

The following text has been added to add this information:

 *They consisted in eddy-covariance systems equipped with a gas analyser to measure the turbulent fluxes and buried sensors including buried temperature sensors and flux plates, allowing the measurement of the ground flux directly and correction of its measurement to the surface (see Table E1) with the calorimetric method (de Silans et al.*
*1997).*

**Table E1.** Instruments at the selected SEB stations (N.A. represents non-applicable). CSAT3, EC150 and Krypton are Campbell Scientific (CS) Instruments; LI stands for LI-COR, HS-50 and R3-50 are Gill models, CNR1, CNR4, CRg4 and CM21 are Kipp & Zonen devices, the flux plates are made by Hukseflux. There were two SEB stations at La Cendrosa from the CNRM and from WUR.

| Site | Sonic and gas analyzer height (m) | Net radiation, height (m) | Flux plate depth (cm) | Soil temperature: depths (cm) | soil moisture depths(cm) |
|---|---|---|---|---|---|
| La Cendrosa | Gill R3-50, LI7550, 3.0 | CNR4, 1.0 | 3 | Generic Pt100, 5, 10, 30 | Delta T Thetaprobe ML3, 5, 10, 30 |
| La Cendrosa | IRGASON, LiCor7500, 1.0 | CM11 & CG2 | 5 | Generic Pt100, 2, 10 | N.A. |
| Els Plans | HS-50, Krypton, 2.0 | CRg4/ CM21, 1.0 | 2 | Delta T ST2 1, 4, 10, 17, 35, 50 | Delta T ML3, 10, 20, 30, 40 |

6. Section 3.3, line 288. The fact that the albedo is lower in El Plans than in La Cendrosa cannot explain the lower value for the Rn. Low albedos increase net radiative heat fluxes.

You are right, and furthermore the albedo is quite variable at this site, so depending on the period, it is lower or higher, which it is explained later in its own section. We now comment on the time difference. The sentence has been changed to:

"The Els Plans site is a rainfed, relatively dry area with natural grass that was drying during the LOP. The parcel is located within a special protection area for steppe birds, and it is not cultivated. The energy budget of a short dry-down period near the end of the LOP (Fig. 3) *with two small rain contributions of 2 mm and 0.8 mm on the eve of 2 September*. The Rn difference is due to the because the contribution of the net long wavelength radiation is lower and time difference between the two periods which can account up to 100 W m-2."

The additional rain sentence is due to another reviewer's comment.

7. Section 4.1, line 305. "The wind speed was filled with the 3 m data at the same site", It is the first time that observations are reported at this height in the manuscript. Was the wind speed recorded at 10 m or 3 m above ground? or at both heights?

It was measured at both heights. The text now reads as:
"The wind speed was filled with an additional wind measurement at 3 m at the same site, corrected by adjusting the wind speed to that which was already measured at 10 m above the surface assuming a logarithmic profile at neutral conditions."

8. Section 4.2.1. Is this large section needed? A summary with a table showing the values used in the runs should be sufficient. Results section starts at page 17!!!

The section has been reduced to a parameter value selection and its full version is sent to an appendix. While we agree with this reviewer (about removal of this text from the main body), we

believe it is important to maintain it in the document as its contents are not reported in recent literature and fill a gap in knowledge that helps setting up simulations of this kind.

9. Section 5.1. Why do you think the ground heat flux is not improved with the new dry surface layer approach?

The heat is not being transferred to the atmosphere but to the soil. The roughness length can be increased to reduce the effect but the value of LE is then significantly reduced. A choice of configuration has been made for the purpose of this article. Some clarification has been added: "After irrigation, the NON simulation better reproduces the soil temperature diurnal pattern, the DSL simulation increases the temperature at 5 cm up to 5ºC in response to the increase in G due to the decrease in LE. *The interaction between the atmosphere and the ground is insufficient and heat is stored instead of being transformed into sensible heat. While the roughness length could be increased to reduce this effect, the characterization of LE would be impacted negatively*" The subsection 'Parameter selection' justifies the choice of roughness length, which is taken within the tabulated values in the literature.
How is this term calculated in the LSM?

This term is extracted by the residual energy not spent in LE and H. It is presented as the ground flux at the surface.
A brief description of its calculation has been added to an appendix. The text reads as:
*The energy budget equation on the SURFEX model can be written as:*

$$Rn - H - LE - G' - S = 0$$

*where S is the energy storage term given by the transfer of energy and G' the ground flux given a particular ground level. G is given then by G' - S and it is defined at the surface.*

We refer to it in section 2.1 with the sentence:
*For completion, the G flux is also described in Appendix C1.*

10. Section 5.1, line 476. I was not able to find in Table 6 the daily RMSE reported for the ground heat flux in the manuscript (i.e., 57 W/m2).

The value should be 70 W m-2, it was erroneously taken as the value of La Cendrosa. It has been modified in the revised manuscript. The correlation values in the following text have been also corrected.

11. Figs 4-5. The period analyzed in these figures should be included in the captions.

The caption has been modified as follows. Figure 4 caption now reads as:

Scatterplots of the simulated terms of the energy budget against the observation for La Cendrosa site for the NON simulation (NON, a-d) and the DSL simulation (DSL,e-h). From left to right, Rn (a,e), H (b,f), LE (c,g), and G (d,h). *The simulation period is from July 1 at 00 UTC to August 1 at 00 UTC.*

Figure 5 caption now reads as:

Scatterplots of the simulated terms of the energy budget against the observation for Els Plans site for the simulation with no resistance (NON, a-d) and simulation with the DSL approach (DSL,e-h). From left to right, Rn (a,e), H (b,f), LE (c,g), and G (d,h). *The simulation period is from June 17 10UTC to September 29 at 9UTC.*

12. Figs. 6, 7, and 8. The format of the days simulated (DD/MM) is not adequate.

It has been changed to MM/DD as in Figure 2 and 3.

13. Fig. 7b. Why does Fig. 7b show results for three months and Fig. 6b only shows results for 15 days for the Els Plans site?

Length of the series has been decided for viewing purposes, to better identify LE changes. VWC can be easily seen for a long period but flux series (see below) need shorter periods to be able to identify the processes.

[Figure]

14. Section 5.2.1. The resistance values shown in Fig. 7 are very different between S92 and DSL simulations, why do you think these significant differences are not producing substantial latent heat fluxes disparities?

Because there is not sufficient water. At some point no more evaporation is happening. We have added the sentence in italics to the text:

The resistance values estimated for Els Plans are similar to those found in Swenson and Lawrence (2014), while those of La Cendrosa are higher, due to the difference in soil properties. The increase in resistance starts earlier than observed in laboratory studies (Zhang et al. 2015). Their

values were closer to the S92 simulation, but slightly higher and remained lower than those shown for the DSL simulation. *These differences result in limited change in LE as there's little water available in the soil, since the VWC is much lower than the field capacity.* In addition, Balugani et al. (2023) found that a DSL observed under natural conditions can be larger than that measured in a lysimeter, whether in laboratory or field conditions. The higher values for resistance compared to the ones by Zhang et al. (2015) may be explained by the exposition to the atmospheric conditions which will affect ET, soil moisture and soil temperature profile (Balugani et al. 2023). To explore this further, a sensitivity analysis is carried out in the following section to test the optimal parameter configuration.

15. Section 5.3.1. Is this section needed?

Yes, we believe that showing the variability of albedo due to the vegetation change for la Cendrosa and due to soil humidity in Els Plans so they can be seen explicitly is important, providing the full picture. Additionally, the setup of the simulation is reflected within this section.

16. Section 5.3.2, Fig. 10a. It seems to me (based on Fig. 10a) that the VWC is better captured by the NON simulation than by the DSL run, especially after the ~ 11th and before the ~ 24th of July, could you explain why?

The difference in VWC values for this period is due to whether the water is either being used by LE or being kept in the soil. The NON simulation overestimates LE signaling a too large loss of water. This explains why the decline in soil moisture is stronger for the non-DSL than for the DSL. Taking into account the tendency of the observations, the 'desired' behaviour is for the water to be evaporated.

While the absolute value is better in the NON simulation for this period it does not describe the processes better. The DSL simulation follows the tendency of the observations better, without crossing the observation, and final loss of water only differs by 1 mm. While being capable of doing both would be preferable, the good modeling of the tendency is prioritary as it is directly related to the evaporation flux. The absolute value is determined in part by the values of the soil field capacity and the saturation value. The problem for both simulations is that field capacity is much larger in the observations than that computed from the pedotransfer equations. As a consequence, infiltration is not sufficient to characterize better the VWC (this is already mentioned in the text). If field capacity is lowered, LE is not well captured.

In summary, we have added the italics text, for clarification, so that the text now reads as :

' ….The simulation has a positive bias of 5\% during *irrigation* flood events caused by a lack of drainage, but the model is able to capture the tendency of the VWC for the DSL simulation. Note that for the NON simulation, this tendency is underestimated. *Although the absolute value is closer to observation for the period before July 24, the rate of loss of water has to be correct for proper LE estimation, and it is thus prioritized over absolute value, which is driven by the field capacity value and which has been calculated with soil samples with CH78 pedotransfer functions. Furthermore, the values of VWC probes can be subject to biases due to the type of probe and manufacturing (Jäckish et al., 2020). The rapid response of the water content at all three*

*observation depths following the flood irrigation event probably indicates that some of the water is being transferred to deeper soil levels via preferential flow pathways in macropores (Nimmo et al., 2025). Implementing a dual permeability approach, as described by Gerke and van Genuchten (1993), could improve the simulation of water flow using the Richards equation in fractures (macropores) and the matrix (micropores) in the future but such changes still face challenges for the implementation at larger scales.'*

For more context, the linkages between surface fluxes and soil properties still has to progress, which is similar to results from Aouade et al (2020). We chose to address the VWC but other studies are working on the pedotransfer improvement (Sobaga et al. 2023) and may be later implemented. A larger issue is that the suitability of pedotransfer functions at different scales is not assured (Weber, et al. 2024). In our case for individual sites, problems can arise from parent material, vegetation and land use. To better capture soil water retention curve, there would be the need to use in addition pressure head observation to capture not only the single time series, but also the pF-curve which was not within the scope of this study.

17. Section 5.3.2, Fig. 10. The format of the days simulated (DD/MM) is not adequate.

We have changed the time format to MM/DD

18. Section 5.3.2, Fig. 10b. Again, the VWC seems better simulated by the NON experiment than by the DSL simulation, could you explain why?

Here, the results of the DSL for latent heat flux were positive but more moderate as the resistance was too strong, and they were dependent on the period. This is observed also here (Fig. 10b) and it is one of the reasons why we later do a sensitivity test to improve the DSL simulation. The loss of water is still better characterized in the DSL simulation, particularly in the two first periods, as Figure 8b indicates with the final accumulated water difference of 1mm. As mentioned previously, parameterisation of soil hydraulic properties, high stone content, sensor bias in absolute values, and macropore existence (see image below) may lead to significant deviations from observations in simulated soil water content when using the NON or DSL approach. Such properties and defects are not readily available on large scales and have not yet become a priority in land surface modelling and are out of scope of the current article.

[Figure]

The text is modified (in italics) as follows:

For Els Plans (Fig. 10b), the same bias is observed in terms of the trend. *The absolute value of the NON simulation being closer to observation for 5 and 10 cm, but deviates significantly from the observation for 30 cm. Here, the DSL appears to capture the redistribution of water following a rainfall event more accurately than the NON, as too much water is used in this approach for soil evaporation. The tendency is larger for the NON simulation than for the observation. For the DSL the tendency is too strong on the wetting events, following what was observed for LE, and giving further necessity for a sensitivity test of the resistance.* The difference between the simulations becomes increasingly larger for each soil water content layer, which decreases after rain events.

19. Section 5.3.3. Fig. 11a shows that the NON simulation reproduces considerably better the maximum soil temperature at 5 cm (below the ground surface) than the DSL simulation for an approximately a 10-day period in the middle of the month (July), could you explain why? Also, the DD/MM format is not adequate here.

A sentence has been added in italics for clarification:

" After irrigation, the NON simulation better reproduces the soil temperature diurnal pattern, and the DSL simulation increases the temperature at 5 cm up to 5ºC in response to the increase in G due to the decrease in LE. *Interaction between the atmosphere and the ground is insufficient and heat is stored instead of being transformed into sensible heat. While the roughness length could be increased to reduce this effect, the characterization of LE would be impacted negatively.*"

We have changed the time format to MM/DD.

20. Section 5.4, line 661. How were the RMSE values of 4 and 2 W m-2 reported here calculated?

Each simulation pertaining to the sensitivity analysis has a RMSE value shown in the colorbar of figure 12a and b. We make the difference against the RMSE value of the DSL simulation of the previous sections for each site compared to the optimal one proposed in the text.   The sentence has been extended to:
The RMSE value which compares the default DSL simulation shown in the previous section to the optimal value is very low, between 4 W m-2 for La Cendrosa and 2 W m-2 for Els Plans.

Bibliography

Aouade, G., Jarlan, L., Ezzahar, J., Er-Raki, S., Napoly, A., Benkaddour, A., Khabba, S., Boulet, G., Garrigues, S., Chehbouni, A., et al.: Evapotranspiration partition using the multiple energy balance version of the ISBA-Ag s land surface model over two irrigated crops in a semi-arid Mediterranean region (Marrakech, Morocco), Hydrology and Earth System Sciences, 24, 3789–3814, 2020.

Balugani, E., Lubczynski, M., and Metselaar, K.: Lysimeter and in situ field experiments to study soil evaporation through a dry soil layer under semi-arid climate, Water Resources Research, 59, e2022WR033 878, 2023

Boone, A., Bellvert, J., Best, M., Brooke, J. K., Canut-Rocafort, G., Cuxart, J., Hartogensis, O., Le Moigne, P., Miró, J. R., Polcher, J., Price, J., Quintana Seguí, P., Bech, J., Bezombes, Y., Branch, O., Cristóbal, J., Dassas, K., Fanise, P., Gibert, F., Goulas, Y., Groh, J., Hanus, J., Hmimina, G., Jarlan, L., Kim, E., Le Dantec, V., Le Page, M., Lohou, F., Lothon, M., Mangan, M. R., Martí, B., Martínez-Villagrasa, D., McGregor, J., Kerr-Munslow, A., Ouaadi, N., Philibert, A., Quiros-Vargas, J., Rascher, U., Siegmann, B., Udina, M., Vial, A., Wrenger, B., Wulfmeyer, V., and Zribi, M.: The Land Surface Interactions with the Atmosphere over the Iberian Semi-Arid Environment (LIAISE) Field Campaign, Journal of the European Meteorological Society, 2, 100 007, 2025.

Brooke, J., Best, M., Lock, A., Osborne, S., Price, J., Cuxart, J., Boone, A., Canut-Rocafort, G., Hartogensis, O., and Roy, A.: Irrigation875

contrasts through the morning transition, Quarterly Journal of the Royal Meteorological Society, 2023.

Canut, G.: LIAISE_LA-CENDROSA_CNRM_MTO-FLUX-30MINL2[Dataset]Aeris, https://doi.org/10.25326/320, 2022a.

Gerke, H. H., & Van Genuchten, M. T. (1993). A dual-porosity model for simulating the preferential movement of water and solutes in structured porous media. *Water resources research*, *29*(2), 305-319.

Nimmo, J. R., Wiekenkamp, I., Araki, R., Groh, J., Singh, N. K., Crompton, O., ... & Sprenger, M. (2025). Identifying preferential flow from soil moisture time series: Review of methodologies. *Vadose Zone Journal*, *24*(2), e70017.

Sobaga, A., Decharme, B., Habets, F., Delire, C., Enjelvin, N., Redon, P.-O., Faure-Catteloin, P., and Le Moigne, P.: Assessment of the interactions between soil–biosphere–atmosphere (ISBA) land surface model soil hydrology, using four closed-form soil water relationships and several lysimeters, Hydrology and Earth System Sciences, 27, 2437–2461, 2023.

Swenson, S. and Lawrence, D.: Assessing a dry surface layer-based soil resistance parameterization for the Community Land Model using GRACE and FLUXNET-MTE data, Journal of Geophysical Research: Atmospheres, 119, 10–299, 2014.

Weber, T. K. D., Weihermüller, L., Nemes, A., Bechtold, M., Degré, A., Diamantopoulos, E., Fatichi, S., Filipović, V., Gupta, S., Hohenbrink, T. L., Hirmas, D. R., Jackisch, C., de Jong van Lier, Q., Koestel, J., Lehmann, P., Marthews, T. R., Minasny, B., Pagel, H., van der Ploeg, M., Shojaeezadeh, S. A., Svane, S. F., Szabó, B., Vereecken, H., Verhoef, A., Young, M., Zeng, Y., Zhang, Y., and Bonetti, S.: Hydro-pedotransfer functions: a roadmap for future development, Hydrol. Earth Syst. Sci., 28, 3391–3433, https://doi.org/10.5194/hess-28-3391-2024, 2024.

Jackisch, C., Germer, K., Graeff, T., Andrä, I., Schulz, K., Schiedung, M., Haller-Jans, J., Schneider, J., Jaquemotte, J., Helmer, P., Lotz, L., Bauer, A., Hahn, I., Šanda, M., Kumpan, M.,

Dorner, J., de Rooij, G., Wessel-Bothe, S., Kottmann, L., Schittenhelm, S., and Durner, W.: Soil moisture and matric potential – an open field comparison of sensor systems, Earth Syst. Sci. Data, 12, 683–697, https://doi.org/10.5194/essd-12-683-2020, 2020.

de Silans, A. P., Monteny, B. A., and Lhomme, J. P.: The correction of soil heat flux measurements to derive an accurate surface energy balance
by the Bowen ratio method, Journal of Hydrology, 188, 453–465, 1997.

Zhang, C., Li, L., and Lockington, D.: A physically based surface resistance model for evaporation from bare soils, Water Resources Research,
51, 1084–1111, 2015